# Metformin alters mitochondria-related metabolism and enhances human oligodendrocyte function

Nina-Lydia Kazakou[1,5], Nadine Bestard-Cuche[1], Laura J. Wagstaff [1], Kellie Horan [1], Luise Seeker[1,6], Sunniva Bøstrand[1,7], Rana Fetit[1], Rebecca Sherrard Smith[1], Fabio Baldivia Pohl [2], Bjorn Neumann[3,8], Patrick Keeler[4], Robin J. M. Franklin[3,8] & Anna Williams [1] ✉

Metformin rejuvenates adult rat oligodendrocyte progenitor cells (OPCs) allowing more efficient differentiation into oligodendrocytes and improved remyelination, and therefore is of interest as a therapeutic in demyelinating diseases such as multiple sclerosis (MS). Here, we test whether metformin has a similar effect in human stem cell derived-OPCs. We assess how well human monoculture, organoid and chimera model culture systems simulate in vivo adult human oligodendrocytes, finding most close resemblance in the chimera model. Metformin increases myelin proteins and/or sheaths in all models even when human cells remain fetal-like. In the chimera model, metformin leads to increased mitochondrial area both in the human transplanted cells and in the mouse axons with associated increase of mitochondrial function/metabolism transcripts. Human oligodendrocytes from MS brain donors treated premortem with metformin also express similar transcripts. Metformin's brain effect is thus not cell-specific, alters metabolism in part through mitochondrial changes and leads to more myelin production. This bodes well for clinical trials testing metformin for neuroprotection.

Metformin is a guanidine-derivative drug synthesised in the 1920s[1] but was only used from 1957 onwards when it was proven that it could reduce blood glucose levels in type II diabetes mellitus. The mechanism of action of metformin is not fully understood, but one mode of action is via inhibition of Complex I of the electron transport chain in mitochondria, disrupting the AMP:ATP ratio, hence activating AMP-activated protein kinase (AMPK), and promoting catabolism to increase ATP[2]. While metformin remains one of the most commonly used therapeutics for type II diabetes mellitus, there is interest in repurposing metformin as a geroprotective agent in various other diseases[3], especially in the context of neurodegenerative diseases, including multiple sclerosis (MS).

In MS, focal areas of demyelination occur in the brain and spinal cord, depriving neuronal axons of myelin sheaths needed for fast saltatory electrical impulse conduction, and also of metabolic support available through the myelin sheath formed by the parent oligodendrocyte. This leads to clinical deficits via slow nerve conduction and through neuronal/axonal loss. Replacement of the lost myelin sheaths

[1]Centre for Regenerative Medicine, Institute of Regeneration and Repair, MS Society Edinburgh Centre for MS Research, University of Edinburgh, Edinburgh, UK. [2]Laboratory of Molecular Neurobiology, Department of Medical Biochemistry and Biophysics, Karolinska Institutet, Stockholm, Sweden. [3]Wellcome-MRC Cambridge Stem Cell Institute and Department of Clinical Neurosciences, University of Cambridge, Cambridge, UK. [4]Joint Research Center for Computational Biomedicine, RWTH Aachen University, Aachen, Germany. [5]Present address: Novo Nordisk Foundation Center for Stem Cell Medicine (reNEW), University of Copenhagen, Copenhagen, Denmark. [6]Present address: Department of Bioengineering, Stanford University, Stanford, CA, USA. [7]Present address: Department of Medical Genetics, Clinic for Laboratory Medicine, Oslo University Hospital, Oslo, Norway. [8]Present address: Altos Labs - Cambridge Institute of Science, Granta Park, Cambridge, UK. ✉e-mail: anna.williams@ed.ac.uk

by oligodendrocytes in the process of remyelination improves conduction velocity and metabolic support, while increased remyelination measures in humans correlate with less disability in MS[4,5]. Remyelination occurs in MS, but is inefficient, especially with increasing patient age.

One reason why remyelination declines with age is due to the decreased responsiveness of oligodendrocyte progenitor cells (OPCs) to cues that allow them to differentiate into myelin-forming oligodendrocytes and remyelinate the denuded axons[6]. Metformin crosses the blood-brain barrier[7] and treatment promotes functional rejuvenation of aged rat OPCs, restoring their responsiveness to pro-remyelinating molecules and drugs, and subsequently enhances remyelination[8]. Based on this potential pro-remyelination effect, metformin is being tested in clinical trials in MS, singly or in combination with other drugs (e.g., ClinicalTrials.gov IDs: *NCT05893225, NCT05298670, NCT05131828, NCT04121468*) but it is also being trialled as a direct neuroprotective agent in MS *(NCT05349474, NCT06463743 & ISRCTN14048364)* and in other neurodegenerative diseases including Parkinson's and Alzheimer's diseases (e.g., *NCT04511416, NCT04098666, NCT04826692, NCT02733679, NCT05781711)*. Despite these ongoing clinical trials, which are yet to report outcomes, the effect of metformin in human oligodendroglia is uncertain, and human cells show differences to those of rodents. For example, human OPCs express genes that are absent in rodent OPCs[9], transplantation studies show functional oligodendrocyte differences between the species[10–12], and these intrinsic differences may contribute to a greater proportion of subcortical myelinated axons in humans than in rodents[13]. Human OPCs and oligodendrocytes can now be derived from pluripotent stem cells (either embryonic stem cells - hESC or induced pluripotent stem cells – hiPSCs) and grown in vitro relatively easily, using published protocols[14]. However, it is not clear whether these cells accurately recapitulate adult human oligodendroglia when cultured in a monolayer, in organoids or transplanted into immunodeficient mice forming chimeras.

We set out to test this at the transcriptional level with the objective of exploring the effect of metformin in the best available context of relevance to adult humans with neurodegenerative disease. We show that of the three culture systems, hESC-derived oligodendroglia transplanted into immunodeficient *Shiverer* mouse brain share the most transcriptional similarities with adult post-mortem human cells. Furthermore, metformin promotes hESC-oligodendrocyte myelination in this context and leads to both mitochondrial morphological changes in axons and glia as well as transcriptional changes consistent with enhanced metabolic function, of interest for metformin's potential therapeutic effect in providing human neuroprotection.

## Results

### Metformin increases the differentiation of foetal-like hESC-derived monolayer oligodendroglia

Treatment of aged rat OPCs with metformin restores their ability to differentiate into oligodendrocytes in response to pro-differentiation factors[8]. To test whether metformin similarly increases human oligodendrocyte differentiation, we generated hESC-derived OPCs in a monolayer using an established protocol[14] (Fig. 1A, B). After 7 days of metformin treatment, we observed a significant increase in oligodendrocytes, both intermediate (OLIG2$^+$O4$^+$) and more mature (OLIG2$^+$MBP$^+$) oligodendrocytes compared to vehicle (mean increase of $0.70 \pm 0.2$ SEM (fold change-FC) and mean increase of $0.52 \pm 0.23$ SEM FC respectively), similarly to the known pro-oligodendrocyte differentiation drug clemastine fumarate[15,16] (mean increase of 0.58 FC $\pm$ 0.1 SEM and 0.48 $\pm$ 0.17 SEM respectively (Supplementary Fig. S1A and Fig. 1C).

As previously, metformin has been shown to only increase differentiation of aged adult but not young rat cultured OPCs, we decided to investigate whether our hESC-derived monolayer oligodendroglia

had characteristics of adult human cells by single-cell RNA sequencing (scRNAseq) compared with our previously published single-nuclear (sn) RNAseq datasets from adult human postmortem CNS tissue[17,18]. Following quality control (QC) of cDNA libraries, samples and clusters (see "Methods"), our dataset contained 19,462 cells of which 3369 were *OLIG2*-expressing oligodendroglia. Canonical cluster marker interrogation confirmed the presence of other major cell types, such as radial glia (*VIM, NES, HES1, SLC1A3*), outer radial glia (*HOPX, TNC, GLI3*), astrocytes (*AQP4, SPON1*), neurons (*STMN2, DCX*) and cycling cells (*TOP2A, MKI67, PCLAF*) (Supplementary Fig. S1B, C). To closely inspect the oligodendrocyte lineage cells, we subsetted out clusters expressing high levels of the canonical oligodendroglial markers *OLIG1* and *OLIG2* and re-clustered these cells. We identified different clusters of oligodendroglia, using previously described features[17,18], finding three oligodendrocyte clusters (mono_Oligo_1-3), (*SOX10, PPP1R16B, MBP, CNP*): a newly formed oligodendrocyte cluster (*BCAS1, TNFRSF21, LGI3*), an oligodendrocyte cluster expressing more mature markers (e.g., *MAG*) and an astrocyte-like oligodendrocyte cluster (*HOPX, SPARCL1, ID3, VCAM1*). We also found six OPC clusters (*PDGFRA, PTPRZ1*): a major OPC cluster, two cycling progenitor (mono_CyP_1, mono_CyP_2) clusters (*MKI67, PCNA*), a primitive OPC (mono_pri_OPC) cluster (*EGFR, PPP1R14B, DLL1, SOX8*), an oligodendrocyte and astrocyte progenitor (mono_OAPC) cluster (*HOPX, SPARCL1, ID3, VCAM1*) and a metabolically active oligodendroglia progenitor cell (mono_MAO) cluster implicated in regulation of mRNA metabolic processes and stability by gene ontology (GO) analysis (Supplementary Fig. S1D). In addition, we found a committed oligodendrocyte progenitor cell (COP) (mono_COP) cluster (*NKX2-2*) (Fig. 1D, E).

To discover whether any of these cell types were similar to adult tissue types, and since our hESC-derived oligodendroglia were primarily patterned towards a caudal character, we next integrated our data with our previously published snRNAseq adult human spinal cord dataset[18] using canonical correlation analysis (CCA) followed by clustering with Seurat[19]. While hESC-derived OPCs mapped closely to the adult human OPCs, only a few hESC-derived mature-like oligodendrocytes mapped onto adult oligodendrocytes, and these were mainly immature (Fig. 1F). To enable a broader and more unbiased identification of cell types compared with inference based purely on marker genes, we also used machine learning to train an artificial neural network (ANN)[20] to recognise single-cell gene expression profiles of human adult cells and predict the identities of hESC-derived cells, using the expression data of the union of variable features between our *in* vitro-derived dataset and our adult human dataset[18] (420 genes) (Fig. 1G and Supplementary Fig. S2A–D). Early progenitor signatures found in the monolayers were not present in the adult dataset, perhaps as expected, but OPCs again showed similarities between datasets. Monolayer oligodendrocytes demonstrated most similarity with COPs and Oligo_F, an oligodendrocyte population with spinal cord selectivity[18], suggesting that in vitro-derived oligodendroglia are immature when compared to their adult human counterparts. To test this hypothesis, we instead integrated our hESC-derived oligodendroglia data with prenatal OPCs (3593) subsetted from a publicly available snRNAseq dataset[21] from 106 donor brains of 13–40-week-old human foetuses (which developmentally have no oligodendrocytes in the first two trimesters but some emerge in the third trimester) and found that they mapped either onto or very close to each other (Fig. 1H). Most of our hESC-derived oligodendroglia still expressed *SOX2* transcripts, a canonical marker of ES cells but also of the initial stages of human gliogenesis[22] (Supplementary Fig. S3A). Co-expression of OLIG2 and SOX2 at the protein level was also shown by immunocytochemistry analysis of monolayer oligodendroglia at DIV 70 (Supplementary Fig. S3B).

Thus, although they respond to metformin, hESC-derived monolayer oligodendroglia are more similar to foetal than adult oligodendroglia, with heterogeneity corresponding to classical stages of

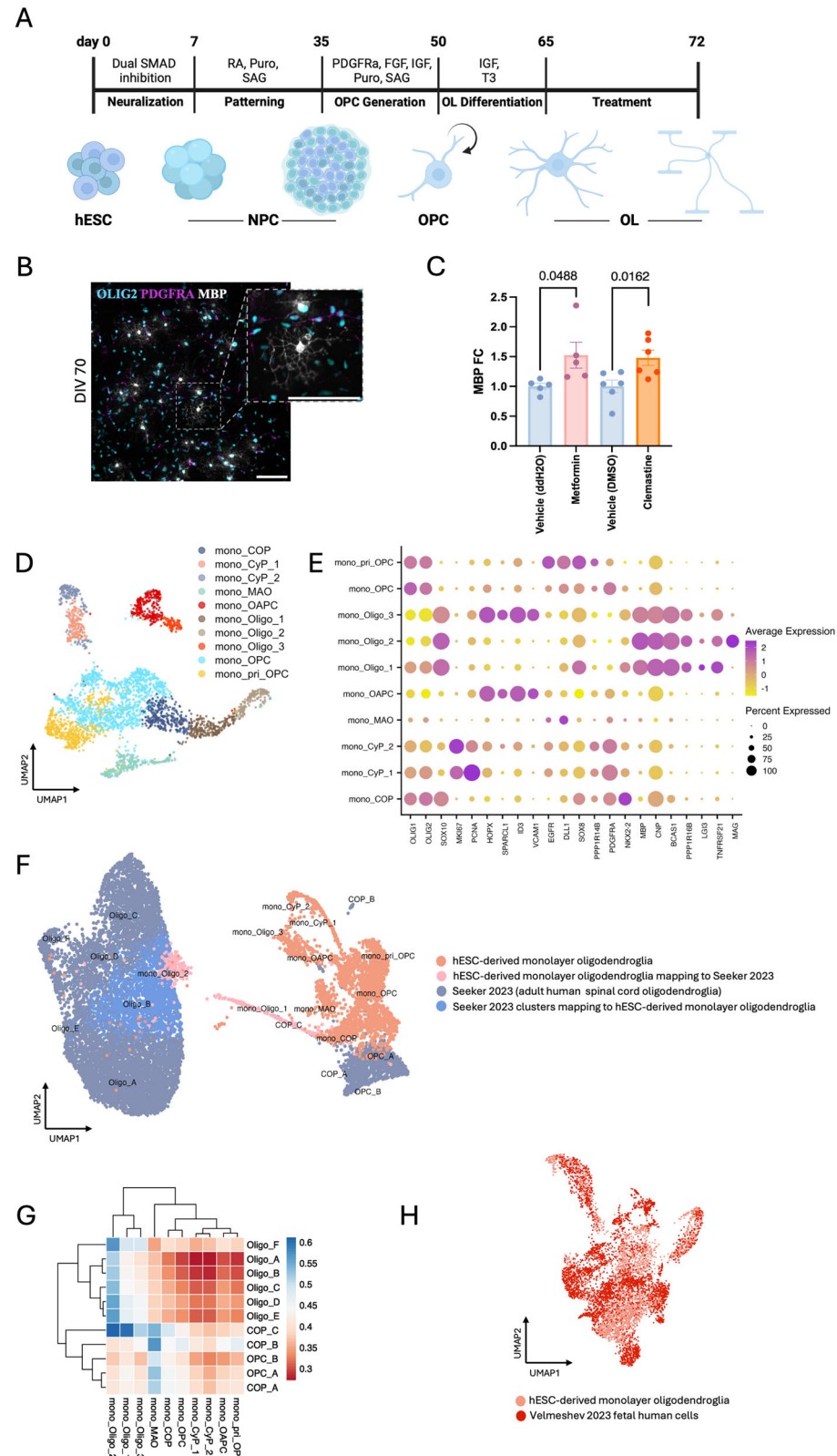

oligodendroglial differentiation. We next hypothesised that organoid cultures, as a self-organised model that allows for interactions among different cell types, including myelination, may contain cells that more resemble adult oligodendroglia, and may be a more appropriate system for testing the response to metformin.

## Metformin increases differentiation of mature human oligodendrocytes in organoids

To identify the effect of metformin on organoids, we established hESC-derived cortical organoid cultures by adapting a previously published protocol[23] (Fig. 2A, B). We treated the cultures with metformin for 10

**Fig. 1 | Metformin treatment increases MBP⁺ human embryonic stem cell (hESC)-derived monolayer oligodendroglia. A** Schematic representation of hESC-derived monolayer oligodendroglia differentiation protocol. **B** Illustrative immunofluorescence image of hESC-derived oligodendroglia at day 70 of in vitro differentiation (DIV) showing PDGFRα⁺ oligodendrocyte precursor cells (OPCs) (magenta) and MBP⁺ oligodendrocytes (grey) co-expressing OLIG2 (cyan). Scale bar = 100 μm. **C** Fold change (FC) difference of OLIG2⁺MBP⁺ oligodendrocytes after treatment with metformin or clemastine compared to their respective vehicle-treated controls (ddH2O or DMSO). *n* = 5 for metformin and vehicle control, *n* = 6 for clemastine and vehicle control where *n* = number of differentiations with 4

technical repeats. Kolmogorov-Smirnov normality test with Dallal-Wilkinson-Lillie for *p*-value, two-tailed unpaired *t* test. Mean ± SEM. **D** UMAP representation of clustered oligodendroglia. **E** Dot plot of selected marker genes showing cluster segregation. **F** Integration with adult human post-mortem spinal cord oligodendroglia snRNAseq dataset[18]. **G** Cosine similarity heatmap showing similarities between the hESC-derived monolayer oligodendroglia (target, labelled mono_) dataset and the adult human spinal cord oligodendroglia (source) dataset. **H** Integration with the second and third trimester foetal OPC snRNAseq dataset[21]. Source data are provided in the Source Data file. Created in BioRender. Swire, M. (2025) https://BioRender.com/j633z18.

days, from DIV 60–70, and assessed oligodendrocyte maturation at DIV 100. Compared to vehicle controls, although the number of either CC1⁺ or MBP⁺ oligodendrocytes did not change (Supplementary Fig. S4A, B), the area of MBP immunofluorescence was significantly increased following metformin treatment (0.45 ± 0.18 SEM) (Fig. 2C).

To define how similar these oligodendroglia were to human tissue counterparts, we performed scRNAseq and, following QC (as before, see "Methods"), our dataset contained 12,923 cells of which 301 were *OLIG2*-expressing oligodendroglia. Canonical cluster marker interrogation confirmed the presence of other major cell types, such as radial glia (co_RG) (*VIM*), glioblasts (*HESS⁺BCAN⁺GFAP⁺*, *HESS⁺BCAN⁻GFAP⁻*), astrocytes (*GFAP, AQP4*), oligodendroglia (*OLIG1, OLIG2*), cycling cells (*MKI67, TOP2A*), pericyte-like cells (*PDGFRB*) and neurons (*DCX, GAD2, SLC17A7*) (Supplementary Fig. S4C, D). We subsetted for *OLIG2⁺* oligodendroglia, revealing a finer distinction of cellular clusters, and although we identified two oligodendrocyte clusters, three OPC clusters as well as a COP cluster (Supplementary Fig. S4E, F), we decided to combine these profiles with those of a publicly available scRNAseq (Smartseq2) dataset[23] using CCA to increase the numbers. Here, we identified two distinct populations of oligodendrocytes (*SOX10, MBP*): a newly formed oligodendrocyte cluster (*BCAS1, GPR17*) (co_Oligo_1) and an oligodendrocyte cluster expressing more mature markers (*MAG, MOG, MOBP*) (co_Oligo_2). We found four OPC populations (*PDGFRA, PTPRZ1*): a traditional OPC cluster (*PCDH15*) (co_OPC), a primitive OPC (co_pri_OPC) cluster (*EGFR, DLL1, HES6*), an oligodendrocyte and astrocyte progenitor (co_OAPC) cluster (*EGFR, SPARCL1*), and a cycling progenitor (co_CyP) cluster (*MKI67, TOP2A*), as well as a COP cluster (*NKX2-2*) (co_COP) (Fig. 2D, E). Many of these oligodendroglia still expressed *SOX2* transcripts (Supplementary Fig. S3C).

On integration of these data with our previously published snRNAseq adult human primary motor cortex (BA4) dataset[18], as before, OPC populations clustered with or close to adult OPCs, co_Oligo_1 clustered with adult COPs and *MAG⁺* co_Oligo_2 and immature co_Oligo_3 clustered with a subset of adult human cortical oligodendrocytes (Fig. 2F). To further compare the identities of hESC-derived and adult oligodendroglia, we employed the same ANN approach using expression data of the union of variable features (181 genes) of the two datasets (Fig. 2G and Supplementary Fig. S2E–H), demonstrating similarity between co_Oligo_2 and adult oligodendrocytes, co_Oligo_1 and adult COPs, as well as co_OPCs to adult brain tissue OPCs.

Thus, hESC-derived cortical organoid myelin protein-expressing oligodendroglia share similarities with adult tissue COPs and immature oligodendrocytes, with heterogeneity linked to developmental differentiation stages, but they are few in number. Due to this limitation, we next decided to test metformin treatment on hESC-derived OPCs transplanted into immunocompromised *Shiverer (Rag2⁻/⁻:Shi/Shi)* mice. This model allows cells to differentiate in an in vivo environment, but in the absence of compact MBP⁺ myelin, due to the mutation in *Mbp* in this inbred line[24] – a method developed by the Goldman lab[10]. Therefore, any MBP⁺ myelin formed is, by definition, attributable to the human cells.

## Metformin increases the percentage of myelinated axons in the mouse corpus callosum by adult-like hESC-derived oligodendroglia

We generated hESC-derived spheres containing PDGFRA⁺ OPCs[14], dissociated them and transplanted 500,000 cells into the corpus callosum of *Rag2⁻/⁻:Shi/Shi* P2-P4 mice (Fig. 3A). At 70 days post transplantation, a mean of 46.77% ± 4.39 SEM of corpus callosal cells are human, as shown by human nuclear antigen (HuNu) immunostaining-positive cells and a mean of 70.51% ± 2.28 SEM of these were OLIG2⁺, showing oligodendroglial lineage (Supplementary Fig. S5A–D), with compact myelination of a mean of 16.15% ± 1.88 SEM rodent axons (Fig. 3B). Myelinated axons are defined here as those with compact myelin (of human origin) and axons surrounded by uncompacted membranes (likely mouse origin) are grouped with unmyelinated axons. We treated these chimeric animals with metformin daily by oral gavage for 21 days, between days 42–63 post transplantation and assessed compact myelination at 70 days post transplantation by electron microscopy, finding a significant increase in the percentage of myelinated axons (and concomitant decrease in unmyelinated axons) after metformin treatment compared to vehicle control (from 21.44 ± 2.3 SEM to 28.21 ± 1.9 SEM) (Fig. 3C) but no change in the number of mature CC1⁺ oligodendrocytes (Supplementary Fig. S5E, F), consistent with the organoid data (Supplementary Fig. S4A, B). We also observed a significant decrease in the average g-ratio between treated and control groups (from 0.84 ± 0.004 SEM to 0.81 ± 0.009 SEM), independent of axonal diameters, suggesting that treatment with metformin also increased myelin thickness (Supplementary Fig. S5G, H).

To identify the fidelity of these hESC-derived cells to adult tissue, we dissected out the corpus callosum of chimeric mice, carried out scRNAseq of mixed mouse and human origin cells and separated them into species computationally. Following QC, our human dataset contained 2235 cells, of which 69 cells were astrocytes (*AQP4, GJA1*) and the rest *OLIG2*-expressing oligodendroglia. We identified five human oligodendrocyte clusters (*SOX10, MBP*) with two mature oligodendrocyte clusters (*MAG, NFASC*): chi_Oligo_5 (*MOBP, OPALIN*) and chi_Oligo_4 (*KANK2*), and three newly formed human oligodendrocyte clusters (*GPR17, BCAS1, TNR, SMOC1*): chi_Oligo_3 (*GRIA2*), chi_Oligo_2 (*SIRT, TMEM108*) and chi_Oligo_1 (*UPC2*). We also identified three human OPC clusters (*PDGFRA*): a major OPC cluster (*PCDH15, PTPRZ1*) and two cycling progenitor (chi_CYP_1, chi_CYP_2) clusters (*MKI67, TOP2A*), as well as a human COP cluster (*FABP7, GAP43*) (Fig. 3D, E). Of note, the protein phosphatase 1 regulatory subunit 16B (*PPP1R16B*) acted as a marker of hESC-derived oligodendrocytes in the monolayer, organoid and chimera models. Many of these oligodendroglia still expressed *SOX2* transcripts, (Supplementary Fig. S3D).

Integration of our human data with our previously published snRNAseq dataset of both adult human brain and spinal cord[18] as before, revealed that hESC-derived OPCs clustered with adult OPCs, newly formed oligodendrocytes (chi_Oligo_1, chi_Oligo_3) clustered with adult COPs and both hESC-derived chimeric mature oligodendrocyte clusters (chi_Oligo_4, chi_Oligo_5) and a newly formed oligodendrocyte cluster (chi_Oligo_2) mapped onto adult human

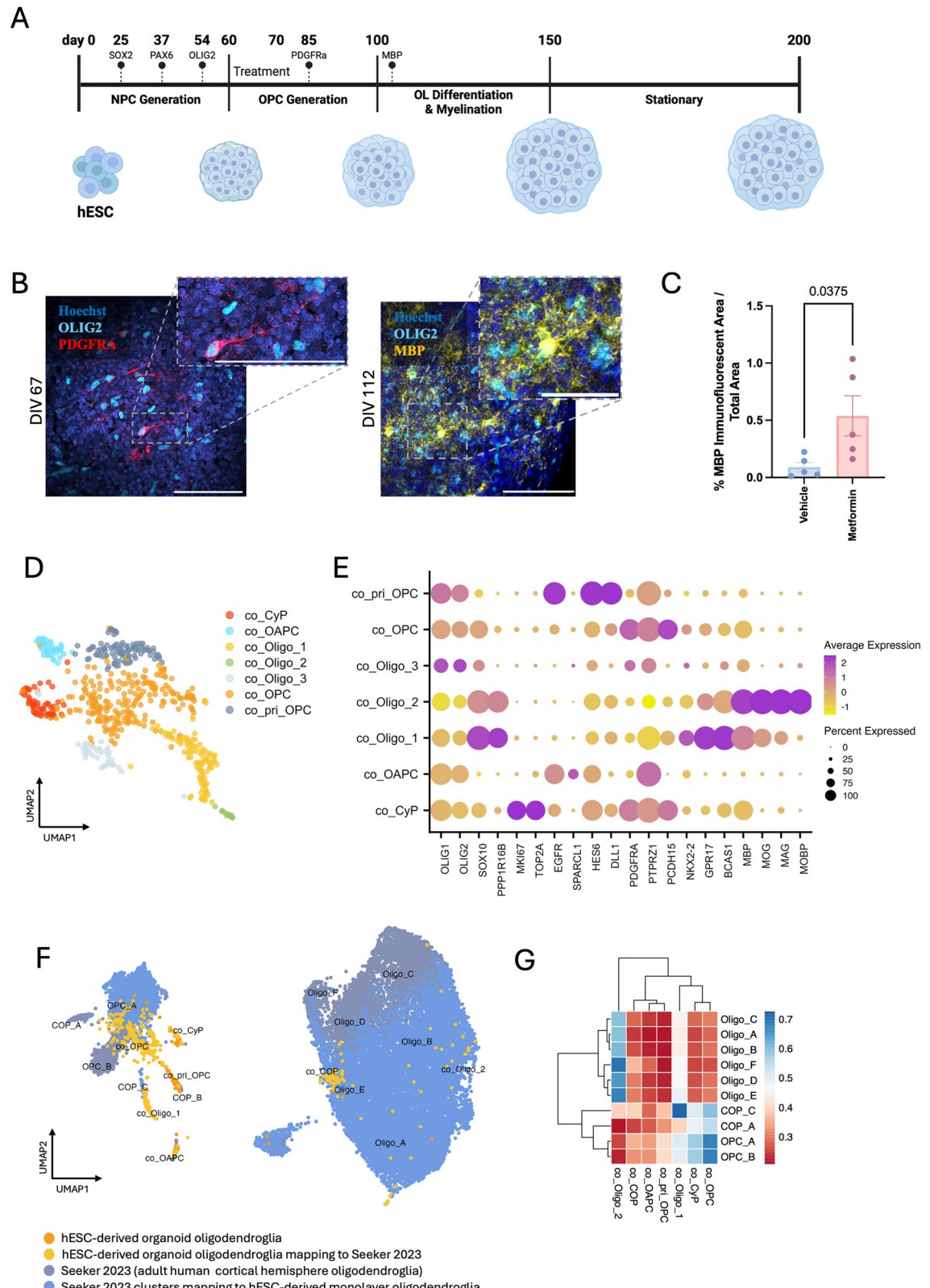

oligodendrocyte subsets from brain and spinal cord combined (Fig. 3F) or separated (Supplementary Fig. S6). ANN analysis using expression data of the union of variable features (346 genes) of the two datasets confirmed these transcriptional similarities (Fig. 3G and Supplementary Fig. S2I-L).

Thus, as hESC-derived chimeric oligodendroglia appeared most transcriptionally similar to those found in the adult human brain and spinal cord, we investigated the effect of metformin treatment further in this model.

## Metformin increases hESC-derived oligodendroglia differentiation and myelination through mitochondrial alterations

Ultra-structural level analysis of control and metformin-treated animals demonstrated obvious differences in the mitochondrial profiles

**Fig. 2 | Metformin treatment increases MBP⁺ human embryonic stem cell (hESC)-derived cortical brain organoid oligodendroglia. A** Schematic representation of hESC-derived cortical brain organoid oligodendroglia differentiation protocol. **B** Illustrative immunofluorescence images of hESC-derived oligodendroglia in organoids at day 67 of in vitro differentiation (DIV), showing PDGFRα⁺ oligodendrocyte precursor cells (OPCs) (red) co-expressing OLIG2 (cyan) and of day 112 showing MBP⁺ oligodendrocytes (yellow) co-expressing OLIG2 (cyan). Scale bars = 100 µm or 50 µm (zoom). **C** MBP⁺ area of immunofluorescence per total organoid area after treatment with metformin compared to vehicle-treated controls (ddH2O). *n* = 5 biological replicates. Kolmogorov-Smirnov normality test with Dallal-Wilkinson-Lillie for *p*-value, two-tailed unpaired *t* test. Mean ± SEM. **D** UMAP representation of clustered organoid-derived oligodendroglia integrated with scRNAseq organoid data from (Marton et al.[23]). **E** Dot plot of selected marker genes showing cluster segregation. **F** Integration with adult human post-mortem cortical hemisphere oligodendroglia snRNAseq dataset[18]. **G** Cosine similarity heatmap showing similarities between the hESC-derived brain organoid oligodendroglia (target, labelled with co_) dataset and the adult human cortical oligodendroglia (source) dataset. Source data are provided in the Source Data file. Created in BioRender. Swire, M. (2025) https://BioRender.com/cl7w2km.

in neuronal axons (whether myelinated or without compact myelin (Fig. 4A, B and Supplementary Fig. S7A) and cytoplasm of glia (Fig. 4C, D), with a significant increase of morphological mitochondrial area after metformin treatment when normalised to the size of axon or cell. *Shi/Shi:Rag2⁻/⁻* mice without transplants do not show this phenotype (Supplementary Fig. S7B). Furthermore, we also analysed available electron micrographs from aged rat brains treated with metformin[8] and identified a similar trend to an increase in mitochondrial profiles in axons (Supplementary Fig. S7C–E), showing that this effect is not particular to these inbred mice. This visible increase in mitochondrial profile area could be attributed to either an increase in mitochondrial number, or their size, or both.

To determine whether the observed mitochondrial morphological alterations had an impact on oligodendroglial function, we next examined our transcriptional data generated as above. There was no marked difference in clusters as seen by compositional analysis between metformin-treated and control groups (Supplementary Fig. S8), but differential gene expression analysis under pseudo-bulk conditions showed 132 upregulated and 54 downregulated genes (average log2FC ≥ 0.5, adjusted pval ≤ 0.05) in oligodendroglia after metformin treatment (Supplementary Data S1). Gene ontology (GO) analysis suggested that these upregulated genes were related to chaperone protein responses, protein folding, and proton/ion transporter activity (Fig. 4E), suggesting a metabolic theme. The most highly expressed specific genes in metformin-treated oligodendrocytes included two nuclear-encoded genes related to mitochondrial function, NADH:ubiquinone oxidoreductase subunit A11 (*NDUFA11*) and cytochrome C oxidase subunit 8 A (*COX8A)*, as well as the eukaryotic translation initiation factor 1 (*EIF1*) (Fig. 4F, G). The *NDUFA11* subunit is required for the assembly of mitochondrial respiratory chain (MRC) Complex I, and *COX8A* is a subunit of MRC Complex IV. *EIF1* was also upregulated in OPCs after metformin treatment and is known to stimulate the translation initiator of short 5′ UTR (*TISU*), enriching mitochondrial mRNAs[25]. We were able to validate similar changes on the chimera tissue using in situ hybridisation RNA probes for *NDUFA11* and *EIF1* (Fig. 4H). Other upregulated transcripts after metformin treatment related to mitochondrial function included *TOMM20* and *CHCHD2* and these were also increased at the protein level after treatment of hES-derived oligodendroglial monocultures for 7 days with metformin (Fig. 4I). TOMM20 is located on the outer mitochondrial membrane and considered as a marker of mitochondrial metabolic activity[26] and CHCHD2 is a transcription factor that regulates mitochondrial dynamics leading to an increase in oxidative phosphorylation[27].

Our strategy of computationally separating the transplanted human oligodendroglia and mouse cells also allowed us to investigate the effect of metformin on mouse oligodendrocytes and other corpus callosal cells (Supplementary Fig. S9A, B). After metformin treatment, we saw an upregulation of *Eif1* (Supplementary Fig. S9B, C), including in mouse corpus callosal oligodendrocytes (average log2FC = 0.7), astrocytes (average log2FC = 0.53), microglia (average log2FC = 0.49) and neurons (average log2FC = 0.3), and *Cox8a* was upregulated in oligodendrocytes (average log2FC = 0.78), astrocytes (average log2FC = 0.5), and microglia (average log2FC = 0.62) (Supplementary Fig. S9B, D). *Ndufa11* was not differentially expressed in mouse

oligodendrocytes or other cells. In addition, *ChChd2* expression is also increased in mouse oligodendrocytes (log2FC > 0.5), and if we use a log2FC threshold of > 0.25, also in mouse neurons, astrocytes and microglia (Supplementary Fig. S9B and Supplementary Data S2–S5). Therefore, the mode of action of metformin is not cell-specific.

We then asked whether there was evidence of activation of these pathways in postmortem brains of MS patients known to have been treated recently before death with metformin. In our open access dataset of snRNAseq from brains of donors with MS and controls[28], we were able to identify two donors with MS on metformin at brain donation and two known to have been without metformin treatment, of similar age and sex matched. We subsetted oligodendroglial nuclei from these donors as before, clustered them with Seurat, and by differential gene expression analysis again found *EIF1* upregulation after metformin treatment when compared both to the two untreated MS donors (average log2FC = 1.26) and to controls without MS from the same dataset (average log2FC = 0.75) (Fig. 4J, Supplementary Fig. S9 and Supplementary Data S6, S7). In this limited dataset, we did not detect *NDUF11A* or *COX8A* transcripts, perhaps related to this being snRNAseq, which selects more for nuclear rather than cytoplasmic transcripts. We also have too few donors with clear metformin-use metadata to determine any association between metformin use in MS, pathology (e.g., remyelination) or clinical course, which will be discovered in clinical trials currently in progress (see introduction). However, even though we acknowledge that our numbers are small, this does suggest a similar mode of action of metformin on oligodendrocytes in the human brain.

## Discussion
Here, we find that human ES-derived oligodendrocytes are foetal-like in monocultures, more mature but few in number in organoid cultures, and transcriptionally most similar to postmortem adult human oligodendrocytes after transplantation into and 'incubation' in mouse brain in vivo as chimeras. Treatment with metformin stimulates human ES-derived oligodendroglia to produce more myelin proteins in all models and more myelin sheaths around axons in the chimeric model. This is different from our previous rat model, where metformin treatment had no effect on young OPCs in monoculture but enhanced the response of aged OPCs to respond to other pro-remyelination drugs[8], although metformin was not tested for its action on remyelination in young rats. However, others have demonstrated a positive effect of metformin in demyelination models in young mice[29–32]. Metformin may also have global effects of relevance, including its potential to reduce blood glucose, blood pressure, and increase cerebral blood flow even in the nondiabetic[33,34].

We show that this response to metformin leads to alterations in mitochondrial morphology and function in multiple cells in rodents and humans. There was an increase in mitochondrial profiles in human and mouse glia and in mouse brain axons, whether myelinated or not, in our chimera model, and in aged rat brain axons after metformin treatment, although we cannot determine if this is due to an increase in mitochondrial number or size or both. A compensatory increase in axonal mitochondria numbers is described in response to demyelination (axonal response of mitochondria to demyelination – ARMD)[35,36],

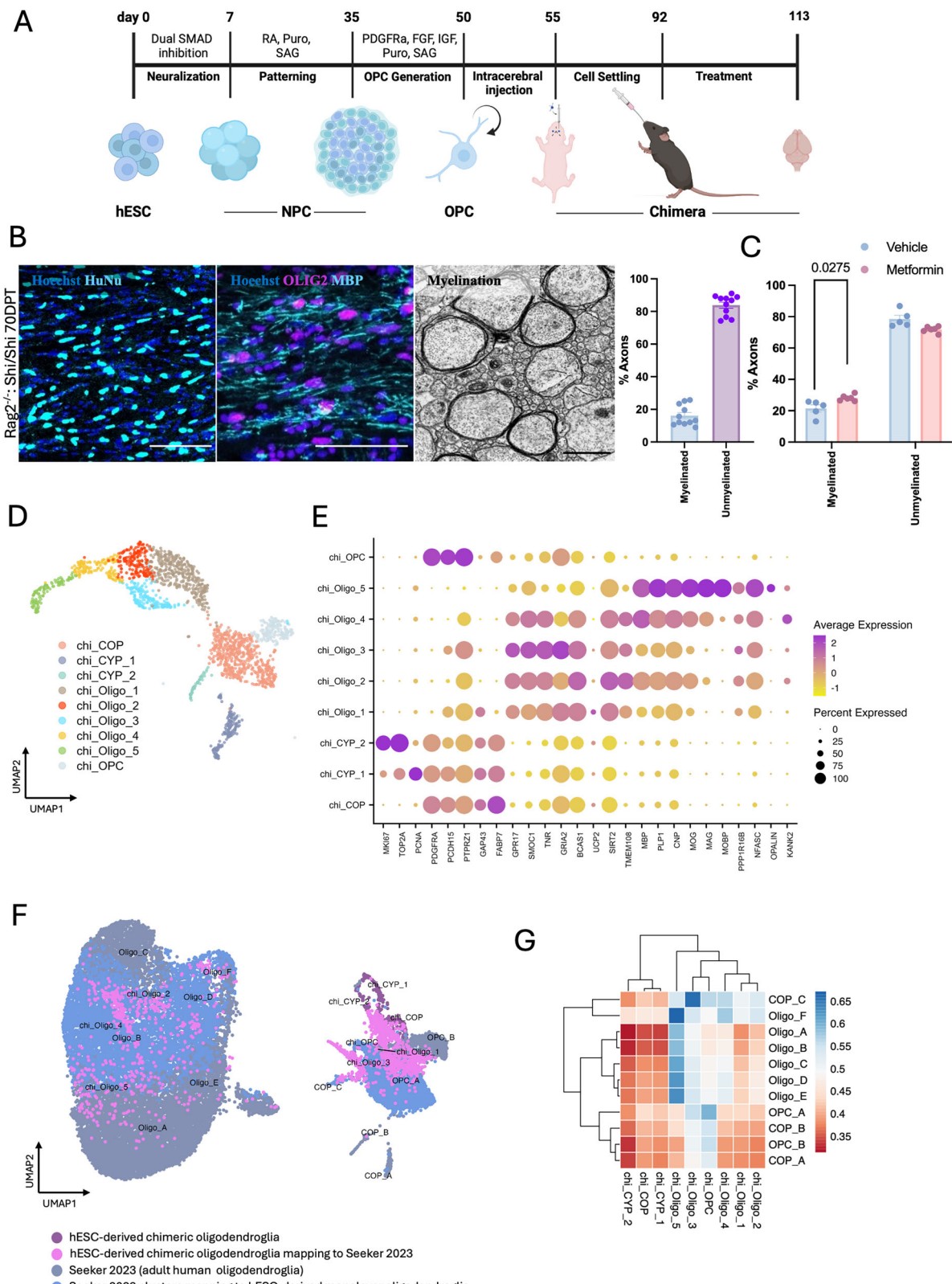

and this axonal mitochondrial number increase can be further boosted by pioglitazone treatment, another drug used in type II diabetes, but working through the PPAR-γ pathway[35]. However, our finding may also be explained by an increase in size or branching of mitochondria, which can occur via mitochondrial fusion, thought to contribute to the enhancement of the mitochondrial networks within cells, improving energy supply and distribution[37]. Mitochondrial structural changes

may also reflect a switch in energy supply utilisation. These changes in both axons and glia emphasise a generic mitochondrial dynamic response independent of compact myelination and may be important to maintain their exquisite interactions.

Our transcriptomic data from human and mouse glia provide evidence that these ultrastructural mitochondrial changes are associated with changes in mitochondrial function. There is increased

**Fig. 3 | Metformin treatment increases the percentage of myelinated axons in chimeric mice. A** Schematic representation of green fluorescent protein-positive (GFP+) human embryonic stem cell (hESC)-derived PDGFRA+ oligodendrocyte precursor cells (OPCs) transplantation into the corpus callosum of Rag2-/-: Shi/Shi P2-P4 mice. **B** Illustrative immunofluorescence and electron micrograph (EM) images of chimeric corpus callosum at 70 days post transplantation showing HuNu+ cells (cyan)(left) or MBP+ oligodendrocytes (cyan) co-localising with OLIG2 (magenta)(middle) and myelinated rodent axons (right). Scale bar = 100 μm (fluorescence) or 5 μm (EM). For control untreated mice, a mean of 16.15% ± 1.88 SEM rodent axons were myelinated. *n* = 11 animals. **C** Quantification of myelinated axons showed a significant increase after metformin treatment compared to the

vehicle-treated (ddH2O) controls. Each dot represents an animal, *n* = 5 animals. Kolmogorov-Smirnov normality test with Dallal-Wilkinson-Lillie for *p*-value, two-tailed unpaired *t* test. Mean ± SEM. **D** UMAP representation of clustered oligodendroglia. **E** Dot plot of selected marker genes showing cluster segregation. **F** Integration with adult human post-mortem oligodendroglia snRNAseq dataset (brain and spinal cord)[18]. **G** Cosine similarity heatmap showing similarities between the hESC-derived chimeric oligodendroglia (target, labelled with chi_) dataset and the adult human oligodendroglia (source) dataset. Source data are provided in the Source Data file. Created in BioRender. Swire, M. (2025) https://BioRender.com/ghxthad.

expression of the nuclear-encoded genes *NDUF11A* and *COX8* in oligodendrocytes after metformin treatment, required for the function of the mitochondrial electron transport chain units I and IV, respectively, helping increase ATP production. This remains compatible with metformin's known mechanism of action of inhibition of Complex I, reducing ATP and indirectly activating AMPK, leading to its actions to increase ATP production, via a negative feedback loop. There is also increased expression of *EIF1*, which regulates RNA translational initiation needed for appropriate cellular metabolism via mitochondria, as depletion in a colon cell line reduces ATP and mitochondrial respiration[38]. Our discovery of increased expression of EIF1 transcripts in two donors with MS treated with metformin for diabetes, compared to similar MS donors without metformin treatment, at least suggests similarities in adult human brains.

As myelin proteins/sheaths increase in our models after metformin treatment, we assume that these mitochondrial morphological and transcriptional changes in this experimental context are beneficial. This is supported by cultured young and aged rat OPCs producing increased levels of ATP (as measured by Seahorse) after treatment with metformin[8,32] as well as positive effects on remyelination in response to metformin treatment in other preclinical models of demyelination[29–32,39]. Here, we have not tested metformin in models of MS, as none of these systems involves either demyelination or inflammation. However, the positive compensatory and beneficial action of metformin on multiple cells, in human, mouse and rat, in ways relevant for neuroprotection, is encouraging for the outcomes of the ongoing trials for progressive MS and indeed other neurodegenerative diseases.

## Methods

Our research complies with all relevant ethical regulations: we used human stem cells (RC17 - female) obtained from the UK Stem Cell Bank with UK Stem Cell Authority permission (who hold ethics/consent) and performed animal experiments under project licence PP1335335 (to AW) under UK Home Office Regulations with ethical approval (University of Edinburgh) protocol numbers 2 and 3.

### Human embryonic stem-cells (hESCs) culture

The RC17 hESC line (female) was used under UK Stem Cell Authority permission, previously genetically edited by us to express a GFP construct generating the GFP32 clone line[40]. We checked our cultures for karyotype (normal) and CNVs compared to the reference line using the CytoSNP 850 K BeadChip from Illumina and analysed in GenomeStudio 2.0 with the plug-in cnvPartition 3.2.0 (Supplementary Data S10). Differentiation reagents and media are listed in Supplementary Data S8 and S9, respectively. The hESCs were cultured in iPS-Brew XF (StemMACS, 30-104-368) on biolaminin 521-coated (BioLamina, LN521-05) for monolayer oligodendroglia and in Essential 8 (A1517001, Life Technologies) on vitronectin-coated dishes (A14700, Thermo Fisher Scientific) for cortical brain organoids and passaged at 80% confluency with 0.5 mM EDTA. Cells were incubated at 37 °C, 5% CO2 with daily medium changes.

### Monolayer oligodendroglia differentiation

hESCs were differentiated into oligodendrocytes as described previously[14]. In brief, hESCs were washed with 1x PBS, dissociated with Accutase, and ~1.5 × 10^6 single cells were seeded into an Aggrewell 800 plate (STEMCELL Technologies, no. 34815) in iPS-Brew XF medium supplemented with ROCK inhibitor Y-27632. Cells were incubated at 37 °C with 5% CO2 for 18–24 h, and formed embryo bodies (EB) were transferred into 10 cm dishes. EB were cultured in suspension on an orbital shaker with media changes every other day. For the generation of neural progenitor cells (NPC), EBs were cultured for 7 days with dual-SMAD inhibition in Chemically Defined medium (CDM) supplemented 1 mM N-acetyl cysteine, 10 μM activin inhibitor SB 431542 and 100 μM LDN-193189 (P1 medium). EBs were caudalised by the addition of 1 μM retinoic acid for 7 days (P2 medium). EBs were then ventralised in P3 medium supplemented with 1 μM of SHH pathway smoothened agonist purmorphamine. Ventral spinal cord patterned progenitor cells were expanded by supplementing the medium with 10 ng/ml FGF-2, while subsequent withdrawal of this factor for 14 days caused neuronal and glial differentiation of NPCs. Oligoprogenitor spheres were cultured for a minimum of 14 days and a maximum of 56 days (with chopping occurring every 14 days) in Oligo Base medium supplemented 10 ng/ml FGF-2, 20 ng/ml PDGF-AA, 1 μM purmorphamine, 1 μM SAG, 10 ng/ml IGF-1 and 60 ng/ml T3. Generation of monolayer oligodendrocytes was achieved through papain dissociation of spheres (Worthington Biochemical Corp., LK003150), then seeding onto plates pre-coated with 10% poly-ornithine solution, followed by 1% matrigel, 10 μg/ml mouse laminin and 20 μg/ml fibronectin at ~25,000 cells/0.3 cm². Terminal differentiation of OPCs into oligodendrocytes was achieved by mitogen withdrawal. Cells were cultured for 7 days in Oligo Base medium supplemented with 10 ng/ml IGF-1, 60 ng/ml T3 and 1 × ITS with 50% media changes.

### Cortical brain organoid differentiation

hESCs were differentiated into cortical brain organoids as described previously[23]. In brief, hESCs were washed with 1 x PBS, dissociated with accutase and ~15 × 10³ cells were seeded into 96-well round-bottom ultra-low attachment spheroid microplates in Essential 8 medium supplemented with ROCK inhibitor Y-27632 and incubated at 37 °C, 5% CO2 for 18–24 h, when medium was exchanged for Essential 6 medium supplemented with 2.5 μM dorsomorphamine and 10 μM SB-431542 for dual SMAD inhibition. On DIV 4, medium was further supplemented with 5 μM of the WNT pathway inhibitor IWP-2, until DIV 24. To generate oligodendroglia spheroids, on DIV 6, spheroids were transferred into Organoid Basal medium supplemented with 20 ng/ml EGF and 20 ng/ml FGF-10. Medium changes were performed daily for 10 days and then every other day until day 24. On DIV 12, medium was also supplemented with 1μM purmorphamine, until day 24. On DIV 25, the spheroids were transferred into 24-well ultra-low attachment plates and Organoid Basal medium was supplemented with 60 ng/ml T3, 100 ng/ml biotin, 20 ng/ml NT-3, 20 ng/ml BDNF, 1 μM cyclic adenosine 3′,5′-monophosphate (cAMP), 5 ng/ml HGF, 10 ng/ml IGF and 10 ng/ml PDGF-AA. Medium changes were performed every other day

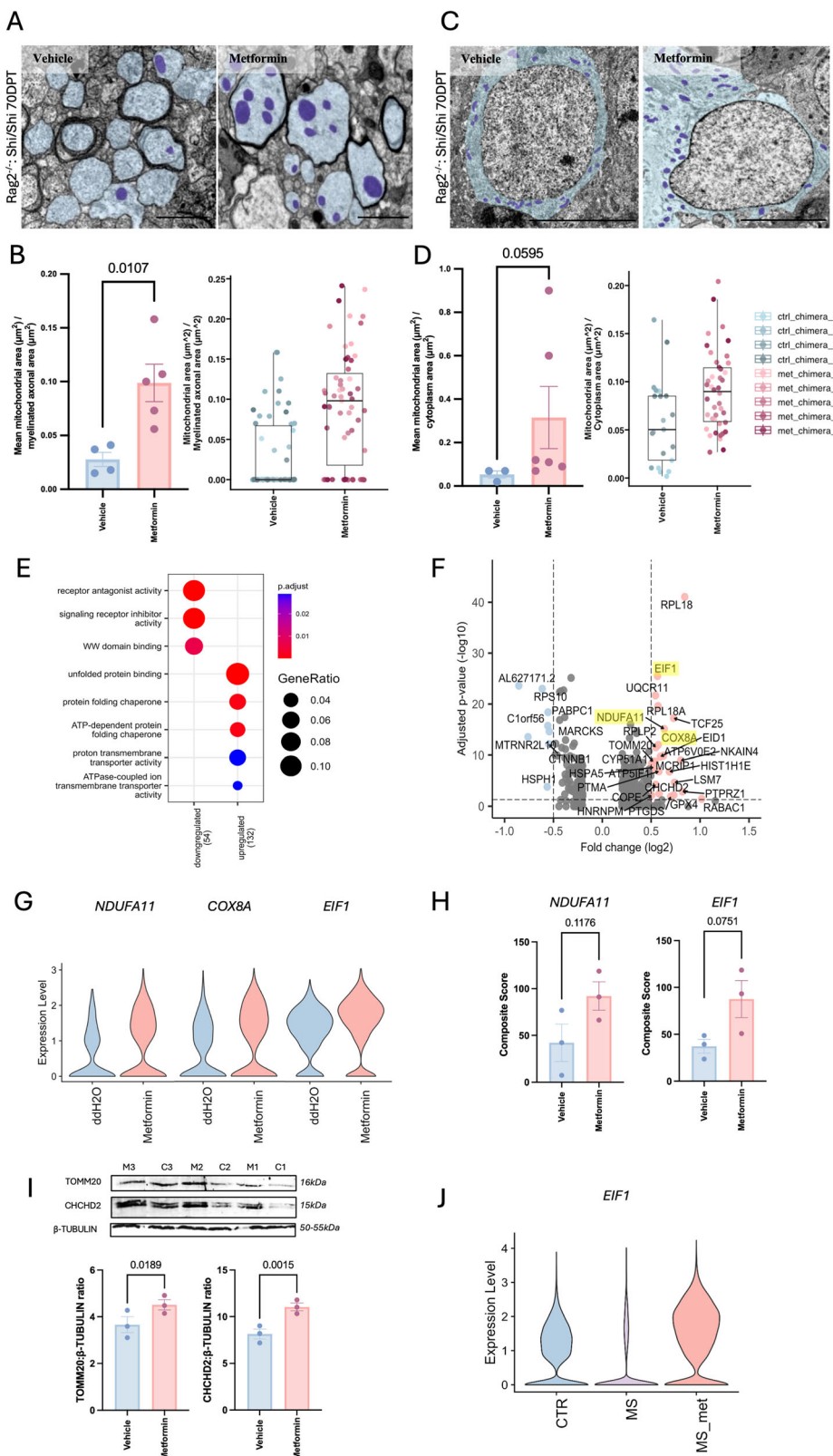

until day 36. From DIV 37 onwards, oligodendrocyte spheroids were cultured in basic medium supplemented with 60 ng/ml T3, 100 ng/ml biotin, 1 µM cAMP and 20 µg/ml ascorbic acid. Medium changes were performed every other day until DIV 43, and from DIV 44 onwards, every 4 days. Spheroids were kept in culture for a maximum of 200 days. Generation of PAX6+ NPC-containing spheroids was

achieved by dual SMAD inhibition, exposure to EGF and FGF10, activation of the SHH pathway and inhibition of the WNT pathway, while treatment with PDGF, HGF, IGF-1, NT-3, BDNF, insulin, T3, biotin, and cAMP led to OLIG2+PDGFRa+ OPC-containing spheroids. Differentiation of MBP+ oligodendrocytes was achieved by the addition of ascorbic acid.

**Fig. 4 | Metformin treatment increases myelination and alters mitochondria.**
**A** Electron microscopy (EM) of myelinated chimeric animal axons showing a significant increase of mitochondrial area after metformin treatment compared to vehicle-treated controls, quantified (**B**) as the average of myelinated axons measured per mouse or number per individual axon measured to indicate the spread. $n = 5$ metformin- and $n = 4$ vehicle-treated animals. Kolmogorov-Smirnov normality test with Dallal-Wilkinson-Lillie for $p$-value, two-tailed unpaired $t$ test. Mean ± SEM. Boxes in plots visualise median and 25th and 75th percentiles, and whiskers mark range up to 1.5 * inter-quartile ranges to show potential outliers. See Supplementary Data S11 for exact summary statistics. Scale bar = 1 μm. **C** EM of chimeric animal corpus callosum showing glia with increased area of mitochondria after metformin treatment compared to vehicle, quantified (**D**) as average of glia measured per mouse or number per individual glial cytoplasm area measured to indicate the spread. Kolmogorov-Smirnov normality test with Dallal-Wilkinson-Lillie for $p$-value, two-tailed unpaired $t$ test. $n = 6$ metformin- and $n = 3$ vehicle-treated animals. Mean ± SEM. Boxes in plots visualise median and 25th and 75th percentiles, and whiskers mark range up to 1.5 * inter-quartile ranges to show potential outliers. See Supplementary Data S11 for exact summary statistics. Scale bar = 1 μm. **E** Gene Ontology (GO) analysis of differentially expressed genes between human embryonic stem cell (hESC)-derived oligodendroglia from metformin- and vehicle-treated chimeric animals. **F** Volcano plot of significantly differentially expressed

genes between hESC-derived oligodendroglia from metformin- and vehicle-treated chimeric animals. pval < 0.05 and logFC > 0.5. DGE analysis was performed using MAST within Seurat, a two-sided statistical test that accounts for detection rate and zero inflation typical of single-cell RNA-seq data. Latent variables to correct for batch effects were included. P-values were adjusted for multiple comparisons using the Benjamini-Hochberg false discovery rate (FDR) method. **G** Violin plot of the most highly expressed specific genes in hESC-derived oligodendrocytes in metformin-treated chimeric animals. **H** Quantification of in situ hybridisation using *NDUFA11* and *EIF1* RNA probes and immunofluorescence for OLIG2 on human cells in chimera tissue with and without metformin treatment. $n = 3$ metformin- and 3 vehicle-treated animals with 2 ROIs of high-density HuNu⁺ nuclei analysed per animal. Composite score = (fraction of OLIG2⁺ cells) x (mean number of RNA probe puncta). Kolmogorov-Smirnov normality test with Dallal-Wilkinson-Lillie for $p$-value, two-tailed unpaired $t$ test. Mean ± SEM. **I** Western blot showing TOMM20 and CHCHD2 levels, compared to housekeeping B-TUBULIN in hESC-derived oligodendroglial monocultures treated for 7 days with metformin versus vehicle, with quantification. Mean ± SEM, two-tailed paired $t$ test. M = metformin treated, C = vehicle treated controls. $n = 3$ separate differentiations. **J** Violin plot showing expression of *EIF1* in multiple sclerosis (MS) donor oligodendrocytes with and without metformin (met) treatment versus controls (CTR). Source data are provided in the Source Data file.

## *Shi/Shi:Rag2*⁻/⁻ OPC transplantations

Animal experiments were performed under project licence PP1335335 (to AW) under UK Home Office Regulations with ethical approval. Homozygous *Shiverer* mice (The Jackson Laboratory, Bar Harbour, ME, C3HeB background) were crossed with homozygous *Rag2*-null immunodeficient mice (The Jackson Laboratory, Bar Harbour, ME, C57/BL6J background) to generate *Shi/Shi:Rag2*⁻/⁻ myelin-deficient, immunodeficient mice. Mice were housed in groups no larger than 5 in individually vented cages (IVCs; Tecniplast, Italy) with unrestricted access to food and water on a 12-h light/dark cycle, at a temperature of 20 °C to 24 °C and humidity of 45% to 65%, in line with the UK's Code of Practice for the Housing and Care of Animals Bred, Supplied or Used for Scientific Purposes. Approximately 500,000 GFP⁺ hESC-derived OPCs were injected using a Hamilton syringe through the skin and skull into both the rostral and caudal corpus callosum of anaesthetised *Shi/Shi:Rag2*⁻/⁻ mice on postnatal day P2-P4. Both male and female mice were used and considered together. Animal numbers indicated in the figure legends.

## Drug treatments

Monocultures were treated every 2 days from DIV 65 to DIV 72 with either clemastine fumarate (1 μM; SML0445-100MG, Scientific Laboratory Supplies Ltd.) or metformin hydrochloride (100 μM; PHR1084-500MG, Scientific Laboratory Supplies Ltd.), or their respective vehicles, DMSO and ddH2O, for 7 days (Fig. 1A). Cortical brain organoids were treated daily between DIV60 to DIV70 with metformin hydrochloride (100 μM; PHR1084-500MG, Scientific Laboratory Supplies Ltd.) or ddH2O for 10 days (Fig. 2A). The organoids were then differentiated further in drug-free medium until DIV 100. Chimeric animals were treated daily with metformin hydrochloride (300 mg/kg; PHR1084-500MG, Scientific Laboratory Supplies Ltd.) or dd2HO by oral gavage for 21days from 92DPT to 113DPT. Treatment was initiated at 42DPT. These doses have previously been used by ourselves[8] and others[41] at timepoints pertinent to affect differentiation.

## Western blotting

Protein samples were prepared by lysing confluent monolayers of human ES-derived ligodendroglial monocultures after 7 days of metformin treatment (as above) in RIPA buffer (Thermo Scientific™ 89900) supplemented with protease and phosphatase inhibitors (Sigma, 535140). Protein concentrations were determined using a Pierce™ BCA Protein Assay (Thermo Scientific™, A55864), and samples

were loaded at 20 μg per lane. Proteins were separated by a 15% SDS-PAGE gel prepared using the Bio-Rad Mini-PROTEAN® system according to the manufacturer's instructions. Electrophoresis was carried out at 150 mV for 1 h. Following separation, proteins were transferred to a 0.45 μm PVDF membrane (Merck, IPVH00010) using a wet transfer system at a constant current of 150 mA for 1 h in cold transfer buffer (25 mM Tris, 192 mM glycine, 20% methanol, pH 8.3). Membranes were blocked in 5% BSA in TBST (20 mM Tris-HCl, pH 7.6, 150 mM NaCl, 0.1% Tween-20) for 1 h at RT. Blots were incubated overnight at 4 °C with anti-TOMM20, CHCHD2 or B-TUBULIN antibodies (see Supplementary Data S9) diluted in 5% BSA-TBST, followed by incubation with fluorescent secondary antibodies at RT for 1 h. Protein bands were visualised and imaged with a ChemiDoc MP Imaging System (Bio-Rad). Densitometry analysis was performed using ImageJ software (NIH).

## Immunocytochemistry

**Monolayer oligodendroglia** were fixed with 4% Formaldehyde (FA) in PBS (v/v) for 10 min at RT, washed in PBS, blocked in 10% heat-inactivated horse serum (HIHS) with 0.1% Triton-X-100 in PBS and incubated at 4 °C overnight with primary antibodies. Cells were washed in PBS and incubated with Life Technologies Alexa Fluor–conjugated secondary antibodies for 2 h at RT. Cells were counterstained with Hoechst and mounted on glass slides using Fluoromount-G™ Mounting Medium (00-4958-02, Thermo Fisher Scientific). Fluorescent images of cells were captured with a Zeiss Axio Observer Inverted microscope. For cell quantification, four 20x fields were imaged from the middle of each coverslip, on three or more different cell differentiations and cell counts were performed manually using the Fiji/ImageJ cell counter plugin.

**Cortical brain organoids** were fixed with 4% FA in PBS (v/v) for 18–24 h at 4 °C, washed with PBS and equilibrated with 30% sucrose for 3 to 7 days. The spheroids were embedded in OCT and sectioned at 12μm. Sections were rinsed in Tris-Buffered Saline (TBS), incubated in pre-boiled 0.01 M citrate buffer pH6.0 for 5 min at 100 °C as antigen retrieval, cooled in running tap water, rinsed in TBS for 2 min and incubated with 3% H₂O₂ for 10 min at RT. The sections were washed with 0.001% Triton-X-100 in TBS (TBS-T), blocked with 10% HIHS with 0.5% in TBS for 1 h at RT and incubated at 4 °C overnight with primary antibodies. The sections were washed in 0.001% TBS-T and incubated with horseradish peroxidase (HRP) (Vector Laboratories) secondary antibodies for 2 h at RT. Stainings were visualised with the Opal Fluorophore Reagent kit (AKOYA Biosciences), washed with 0.001%

TBS-T, counterstained with Hoechst and mounted on glass slides using Aquamount (14-390-5, Thermo Fisher Scientific). For multiplex staining, prior to counterstaining, the sections were incubated with unmasking solution for 2 min at 100 °C and the primary and secondary incubation steps were repeated.

For whole mount spheroid immunohistochemistry, spheroids were fixed in 4% FA in PBS (v/v) for 2 h at RT, washed with PBS, permeabilized with 5% PBS-T for 45 min at RT, blocked with 10% HIHS in 3% PBS-T for 1 h at RT and incubated at 4 °C overnight with primary antibodies in blocking solution. Spheroids were then incubated with Life Technologies Alexa Fluor–conjugated in secondary antibodies for 2 h at RT, counterstained with Hoechst and imaged with a Nikon A1RHD25 confocal microscope.

Automated analysis of MBP quantification was performed on 10-micron-thick cryostat-sectioned spheroids. Sections were immunofluorescently labelled with an antibody against MBP and counterstained with Hoechst. Tile-Scan images of the whole section were acquired with the 10x objective using a Leica SP8 confocal microscope. A custom Fiji/ImageJ macro was used for image quantification. The total tissue section area was defined based on Hoechst staining. The MBP signal was identified by immunofluorescence and reported as the percentage of MBP-positive area relative to the total tissue section area (termed MBP immunofluorescence area).

Quantification of CC1+ MBP+ nuclei on three organoids per condition was performed using ImageJ/Fiji v1.54 f on 40X images taken by the Opera Phenix High-Content screening system.

**10 week-old chimeric mice** underwent intracardiac perfusion-fixation with 4% FA (v/v), dissected brains left in 4% FA in PBS (v/v) for 18–24 h at 4 °C, washed with PBS and equilibrated with 15% sucrose for 18–24 h at 4 °C, followed by 30% sucrose for 1 to 3 days. Brains were then snap-frozen by immersion into dry-ice cold 2-methyl-butane for 15 s and cryosectioned at 16 μm. The sections were rinsed in PBS, blocked in 10% HIHS with 0.1% PBS-T and incubated at 4 °C overnight with primary antibodies in blocking solution. The sections were washed and incubated with Life Technologies Alexa Fluor–conjugated in secondary antibodies for 2 h at RT, counterstained with Hoechst and mounted on glass slides using Fluoromount-G™ Mounting Medium. Fluorescent images were captured with the Vectra Polaris (Perkin Elmer) slide scanner. For MBP+ area quantification, 20x images were collected from each side of the corpus callosum from three animals per condition and analysis was performed automatically using a Fiji/ImageJ macro. Quantification of HuNu+ nuclei and HuNu+OLIG2+ was performed semi-automatically using QuPath v.0.3.0, where a ROI was manually created, the number of nuclei within the ROI was automatically detected and object classifiers were trained to determine single- or double-positive nuclei. Three sections from three to six animals with were analysed. All primary antibodies used are listed at Supplementary Data S9. Quantification of HuNu+ Olig2 + CC1 + was performed in the corpus callosum from three animals per condition using ImageJ/Fiji v1.54 f on 40X images taken by the Opera Phenix High-Content screening system.

**RNAScope In Situ Hybridisation combined with immunofluorescence.** Sections were washed in PBS, blocked in 10% HIHS with 0.1% PBS-T containing mouse-on-mouse blocking reagent (Vector Laboratories, MKB-2213) and incubated at 4 °C overnight with HuNu primary antibody (Millipore, MAB1281). Sections were washed in PBS and incubated with an Alexa Fluor–conjugated secondary antibody (Life Technologies) for 2 h at RT. Sections were counterstained with DAPI (ACD Bioscience, 320858) and mounted on glass slides using Prolong Gold Anti-fade mounting medium (Invitrogen, P36930). Fluorescent images of the sections were captured with the Opera Phenix High-Content Screening System. RNAScope in situ hybridisation was combined with immunofluorescence in sequential sections using the RNAScope Multiplex Fluorescent Reagent kit v2 (ACD

Bioscience, 323100). Sections were washed in PBS, baked at 60 °C for 30 min and post-fixed with 4% PFA at 4 °C for 15 min. Tissue was dehydrated in increasing concentrations of ethanol. Sections underwent $H_2O_2$ treatment for 10 min at RT, washed in ddH2O and incubated for target retrieval for 5 min at 100 °C. Slides were washed again in ddh2O, followed by 0.1% PBS-Tween and incubated with anti-Olig2 primary antibody (Atlas antibodies, HPA003254) overnight at 4 °C. Sections were washed in 0.1% PBS-Tween and post-fixed in 4% PFA at room temperature for 30 min. Sections were washed in 0.1% PBS-Tween before incubation with Protease Plus at 40 °C for 30 min, washed again in ddH2O, and probes were hybridised for 2 h at 40 °C. The following probes were used: Hs-EIF1-No-XMm-C1 and Hs-NDUFA11-No-XMm-C1. Sections were incubated at 40 °C with Amp 1 (30 min), Amp 2 (30 min) and Amp3 (15 min) interspersed with wash buffer steps. Slides were incubated with HRP-C1 for 15 min at 40 °C, followed by Opal dye 570 (Akoya Bioscience, FP1488001KT) for 10 min at 40 °C. Sections were washed with wash buffer, and HRP blocker was added for 15 min at 40 °C. Sections were incubated with an Alexa Fluor–conjugated secondary antibody (Life Technologies) for 45 min at RT Sections were washed with 0.1% PBS-Tween, counterstained with DAPI (ACD Bioscience, 320858) and mounted on glass slides using Prolong Gold Anti-fade mounting medium (Invitrogen, P36930). Fluorescent images of the sections were captured with the Opera Phenix High-Content Screening System. For quantification of mRNA transcripts, 40x images were obtained of the corpus callosum area, and manual quantification was performed using QuPath v0.3.4 and Fiji/ImageJ. Three animals were analysed per probe per condition, with 2 ROIs chosen per animal based on a high density of HuNu+ nuclei in a sequential section. For each probe, the fraction of Olig2+ cells that were positive and the mean number of puncta per all cells were quantified. A composite score was calculated for each probe: (Fraction of Olig2+ cells) x (Mean number of puncta).

**Transmission electron microscopy (TEM).** 10-week-old animals underwent intracardiac perfusion-fixation with 4% FA (v/v) and 2% glutaraldehyde (GA) in 0.1 M phosphate buffer (PB) (v/v), pH = 7.4. Brain extraction was followed by post-fixation in 4% FA in PBS for 18–24 h at 4 °C. 1 mm thick coronal sections were cut using a mouse brain matrix, and the corpus callosum dissected out. The sections were post-fixed in 1% GA in 0.1 M PB (v/v), then in 1% osmium tetroxide for 30 min at RT and processed with an automatic EM Tissue Processor (Leica), as follows: sample dehydration through a series of absolute ethanol washes (50%, 70%, 90%, twice 100% (v/v)) for 10 min each; 100X propylene oxide twice for 15 min; resin & propylene oxide mix (1:1) and resin & propylene oxide mix (2:1), each for 2 h; 100% resin (TAAB 812 Premix Kit-Medium T031) overnight. Samples were then embedded in fresh resin in appropriate moulds and incubated at 60 °C for 16–24 h. Ultra-thin 60 nm thick sections were taken from near the middle of the corpus callosum and stained with uranyl acetate and lead citrate. To determine the percentage of myelinated and unmyelinated axons, ~300–400 axons were scored per animal. Axons with an axon diameter < 0.4 μm were excluded from the analysis, as they are unmyelinated due to their small calibre[42]. For g-ratio analysis, ~20 axons per animal were analysed. The axon area (i) and axon + myelin (ii) area were measured, and the axon diameter (d[i]) and axon + myelin diameter were calculated, assuming circularity (g-ration = diameter(axon area) / diameter(axon + myelin area)). For mitochondrial area quantification, 3 × fields of 5.3 × 5.3 microns containing ~30–35 transverse profiles of axons in total per mouse (n = 5 metformin- and n = 4 vehicle-treated animals) and 7 × fields 10.8 × 10.8 microns containing ~1–2 glia per mouse (n = 6 metformin- and n = 3 vehicle-treated animals) were traced manually and the size calculated using the Fiji/ImageJ TrackEM plugin. Values were analysed both as averages per mouse and as individual measurements per axon/glia. Normality was assessed using the Kolmogorov-Smirnov test with the

Dallal-Wilkinson-Lillie method for *p*-value calculation. Based on normality, statistical comparisons were performed using either a two-tailed unpaired t-test or the Mann-Whitney test.

**Single-cell RNA sequencing.** Monolayer oligodendrocytes were incubated in Accutase solution for 5 min at 37 °C, collected by centrifugation at 400 x *g* for 5 min and resuspended in Oligo Base medium on ice for counting. Organoids were dissociated using the Worthington Papain dissociation system, with 45 min incubation in papain/DNase, pooling ten organoids together per sample. Cells from chimeric animal tissue were collected as previously described[8]. In brief, animals were perfused using ice-cold HBSS[+/+] (14025092, Thermo Fisher Scientific), a brain matrix used to excise the corpus callosum into HBSS[-/-] (12549069, Thermo Fisher Scientific) supplemented with 0.4 units of RNase inhibitors (N808011, Life Technologies) and mechanically dissociated with a scalpel. Using 0.2 units of RNase inhibitors at each subsequent step, the tissue was washed twice with HBSS[-/-], centrifuged for 5 min at 400 x *g*, resuspended in 0.62 units Research Grade Liberase DH (LIBDH-RO, Roche) and incubated for 40 min at 37 °C with gentle rocking. 2 ml of MEM (12492013, Thermo Fisher Scientific) containing 0.5% BSA, 500 units bovine pancreas DNase (DN25, Sigma-Aldrich) and 0.5 ml FBS were added, and the tissue was mechanically triturated with a P1000 pipette and strained through a 70 µm mesh. Tubes containing tissue were filled to full volume with MEM containing 0.5% BSA, centrifuged for 10 min at 400 x *g* and resuspended into 0.2% BSA. Two corpus callosa were pooled together per sample. Live cells were sorted with LSRFortessa analyser (BIO-RAD), and mixed human and mouse cells were centrifuged and resuspended in 100 µl of 0.2% BSA. We processed cell suspensions containing 16,000 viable cells (aiming for recovery of 10,000 cells) by the Chromium Controller (10X Genomics) and constructed barcoded libraries using the Chromium Next GEM Single Cell 3' v3.1 kit (PN-1000121) following the manufacturer's instructions. Samples were sequenced with Illumina NovaSeq6000. For monolayer and organoid datasets, reads were mapped and UMI count matrices were generated with 10X Genomics Cell Ranger Single-Cell Software Suite v5.0.0 using default parameters and reference transcriptome GRCh38 2020-A. For the chimeric oligodendroglia, the STARSolo (STAR version 2.7.8a) was used for sample de-multiplexing, aligning the reads to the GRCh38-and-mm10-2020-A concatenated mouse and human reference genome sequence. The flag --soloMultiMappers was used to include reads that map to multiple loci in the genome, e.g., if the mouse and human genes have a similar sequence, and the flag –outSAMattributes CB was added to include the cell-barcode (CB) to the output BAM files. Cells were allocated to either mouse or human identity based on at least 70% gene mapping to the respective genome. Mice and human CBs were saved for future filtering. To obtain the mice cells, the fastq files were realigned to the mouse (mm10 2020-A) reference genome, using Cell Ranger Single-Cell Software Suite v5.0.0 for consistency with the previous experiments. For human cells, new fastq files containing only the human cell reads were generatedrom the BAM files. Then, the original fastq files were filtered into human-specific fastq files, inputting them and the extracted read IDs to filterbyname.sh from BBMAP[43]. The new human-specific fastq files were aligned using Cell Ranger Single-Cell Software Suite v5.0.0 to the human (GRCh38 2020-A) reference genome. All data analyses were performed using the R software (https://www.r-project.org/). Quality control was carried out using Scater[44]. Genes expressed in ≤10 cells were removed. Cells were filtered out based on high/low UMI counts, high/low gene counts and high percentage of mitochondrial genes. Data normalisation was performed using Scran[44,45]. Principal component analysis (PCA), dimensionality reduction and Uniform Manifold Approximation and Projection (UMAP)[46] were performed using Seurat v4. The most appropriate PC number was determined based on the elbow heuristic, K-nearest neighbour (KNN) graphs were built as a basis for community detection using a Louvain algorithm[47] and initial clustering was performed to resolve broad cell types based on the expression of canonical markers. Sub-clustering of the oligodendroglial lineage was carried out by applying the same approach as before. Differential gene expression analyses for the identification of cluster marker genes and genes altered after treatment with metformin were performed using MAST[48], filtering genes that had a minimum positive log2-fold change (log2FC) of 0.25 and were expressed by at least 20% of cells within the cluster/group of interest and less than 60% outwith. When comparing conditions, additional thresholds of absolute log2FC of ≥0.5 and adjusted *p*-value of < 0.05 were applied.

**Data integration with public datasets.** Publicly available data were retrieved from GEO and integrated with using Seurat's canonical correlation analysis to identify anchors or reciprocal PCA.

**Artificial neural network.** To compare the cell identities of the different models to reference data of human oligodendroglia, we followed a similar approach the one described previously[49]. We trained an artificial neural network (ANN) to recognise single-cell gene expression profiles within a source domain (human cell atlas (HCA)[18]) and used the source-trained ANN to label cells in a target domain of the data reported here. Each model (monolayers, organoids and chimeras) was trained separately. To enhance the learning complexity, we reduced the feature space to the union of variable genes identified in the target and source data (Seurat::FindVariableFeatures). The resulting feature space was binarized to two expression levels: detected (number of UMIs > 0) and undetected (number of UMIs = 0). Based on our previous experience[50], we designed an ANN consisting of an input layer with 3479 units, a first hidden layer with 64 units, followed by ReLU activation function and L1-Regularisation (l = 0.001), a second hidden layer with 32 units, ReLU activation and L1-regularisation (l = 0.001), and a softmax output layer with 28 units, corresponding to the cell type labels provided by the HCA[18]. The training data were split by labels into 5 parts (monolayer & organoid datasets) or 3 parts (chimeric dataset) with the aim to obtain multiple fold cross-validation. The models were trained, each using all parts minus one as training data and the remaining part for validation, for 1000 epochs with a step size of 256, and a sample generator to re-sample 16 training examples per class per step. Loss was calculated using cross-entropy, and gradient descent optimisation was conducted using RMSprop with default parameters. ANNs were implemented and trained using the keras for R package (v2.2.4; https://keras.rstudio.com/) and the TensorFlow backend (v1.8.0; https://www.tensorflow.org/) on a GeForce GTX 1050 GPU (NVIDIA, Santa Clara, CA, USA). Classification performance in the source domain was evaluated using the average balanced accuracy as well as using the average precision and recall, calculated from the validation data held back from training. Classification of cell identities in the target domain was performed from an ensemble of all 3 or 5 ANNs via plurality vote.

**Pathway analysis.** Gene ontology (GO) analysis was performed using cluster profiler[51] on all significantly differentially expressed genes (adjusted *p*-value < 0.05) with a log2FC change of at least 0.5 to identify significantly enriched biological processed in oligodendroglia sub-clusters.

**Statistics and reproducibility.** Data are presented as mean ± SEM unless otherwise indicated. All quantitative data was tested for normality of distribution; statistical analyses were performed using the Student's *t* test (two-sided) or Mann−Whitney U-test (two-sided) for comparison between two groups, all using Graphpad prism. Specific statistical tests are described in all figure legends, and *p*-values reported on the graphs. For clemastine and metformin treatment experiments, samples were randomly distributed between the two

conditions and all data collection and analysis were performed blinded.

Illustrations created with BioRender.com.

All unique/stable reagents generated in this study are available from the lead contact (AW) without restriction upon request.

## Reporting summary

Further information on research design is available in the Nature Portfolio Reporting Summary linked to this article.

## Data availability

Single-cell RNA-seq data have been deposited at GEO GSE282369 and are publicly available. Other human snRNAseq data were obtained from the publicly available datasets: Human MS brain donor data EGAD00001009169, Human adult brain and spinal cord donor data https://cellxgene.cziscience.com/collections/9d63fcf1-5ca0-4006-8d8f-872f3327dbe9, Human foetal brain donor data https://cellxgene.cziscience.com/collections/bacccb91-066d-4453-b70e-59de0b4598cd, Human organoid cultures GSE115011. Source data are provided in this paper.

## Code availability

This paper does not report original code.

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

## Acknowledgements

Funding was from Multiple Sclerosis (MS) Society UK Edinburgh research centre grant (grant code 133)(A.W.), Medical Research Council/ MS Society grant (MR/T015594/1) (A.M., R.J.M.F.), Wellcome Trust Tissue Repair Doctoral training fellowship (N.K.) and postdoctoral funding from F. Hoffmann-La Roche (A.W.). We would also like to acknowledge the help and support of Gonçalo Castelo-Branco, Karolinska Institutet, Matthieu Vermeren at the Centre for Regenerative Medicine Imaging Facility for Fiji/ImageJ macro design, Stephen Mitchell at the Electron Microscopy facility at the King's Buildings campus, University of Edinburgh, for ultra-thin sectioning, Fiona Rossi at the FACS facility at the Centre for Regenerative Medicine for FACS help, and our animal technicians, especially Lorraine McNeil. Thank you to Matt Swire for help with figures.

## Author contributions

Conceptualisation, N.K., R.J.M.F. and A.W.; Methodology, N.K., N.B.C., L.W., K.H. and P.K.; Investigation, N.K., N.B.C., L.W., K.H., F.B.P., R.S.S., R.F., L.S., S.B., P.K. and B.N.; Writing – Original Draft, N.K. and A.W.; Writing – Review & Editing, all authors; Funding Acquisition, R.J.M.F. and A.W.; Resources, B.N. and R.J.M.F.; Supervision, R.J.M.F. and A.W.

## Competing interests

The authors declare no competing interests.
