## [Transparent Peer Review file · Nature Communications]

Metformin alters mitochondria-related metabolism and enhances human oligodendrocyte function

Corresponding Author: Professor Anna Williams

Version 0:

Reviewer comments:

Reviewer #1

(Remarks to the Author)

This paper follows up previous work by members of this group that reported the ability of metformin to enhance the remyelination capability of oligodendrocyte progenitor cells. While that previous study was in rats, the current work seeks to extend that work to humans, to lay the foundation for this approach as a therapeutic strategy. In the present paper, they assessed the effects of metformin in human OPCs in culture, in organoids, and in human glial-implanted mice. The first performed a detailed assessment of the composition of OPCs produced by their differentiation protocol. The authors found that the cells in monolayer or in organoid culture generally retained a fetal-like gene signature, and that the chimera model provided the most adult-like human OPCs, which were their intended targets of metformin. On that basis, they explore the mechanism by which metformin exerted its actions. They found that metformin enhanced myelin gene and protein expression in all three of their models, but that its functional effects were most evident in the chimera model. In that setting, metformin led to increased mitochondrial gene expression and size, that this was noticed in axons as well as oligodendrocytes, and that these changes were also reflected in postmortem human MS tissues derived from patients treated with metformin antemortem. The paper is solid and interesting, a logical extension of the groups' prior studies, and an important preclinical study if the intent is to proceed to clinical trial of this strategy. I do have some concerns though, that the paper would benefit from addressing.

Major points

Figure 1-The author report that a week (7d) of metformin increased the incidence of both O4+ and MBP+ oligodendrocyte stages. Yet the 0.7-fold change they report is underwhelming, and the data are only reported as fold-change; we have no idea what the absolute number and relative percentages are, without which fold-change data are difficult to interpret. In that regard, figure 1B seems to indicate that the MBP percentages are quite low – not surprising, since human OPCs express little MBP in OPC monoculture - typically only with strong T3/IGF1 stimulation unless axons are present - and even then not until longer time points, since human oligodendrocytes take longer to mature in most culture protocols. Overall, these data might have looked better with longer treatment times. It would be helpful if the authors have such data to report, along with the quantitative data – numbers and percentages – upon which their conclusions are based.

Figure 2C- The authors report that in organoids treated for 10 days, that MBP density was significantly increased. But “MBP density” is really qualitative, neither well-described nor strictly quantitative, and again, a fold-increase is reported rather than clear percentages. Also, a correlation with the detailed genomic data, relating the MBP data to the compositional data obtained in the scRNA-Seq analysis (figs. 2D and S2), would be informative.

Figure 3 (page 8)-Again the use of fold-change endpoints seems a bit confusing – the authors report that 16.15% of rodent callosal axons are myelinated in untreated chimeras and that this number increased by 6.76-fold after metformin-treatment – so does that mean that >100% of axon were myelinated? Yet Fig 3C shows a barely perceptible bump in the median percentage of myelinated axons to metformin. I may be misunderstanding the author's argument here, but regardless, this section needs to be clarified, and again, actual numbers and percentages provided.

Figure 4 (page 9) The authors report that both glial and neuronal mitochondrial profiles increased to metformin, by 0.05-0.1-fold depending on size normalization. Yet despite the significance claimed in Fig. 4B, the sample size here seems far too low for a comparison made in 2D TEMs that is significantly confounded by slice orientation. The authors need to make clear

how many cells were sampled in each of the few mice in each group, how many mitochondria/cell, and areas thereof; the fold change endpoint and relatively sparse data shown in 4C are not convincing. The statistics applied need to be better described here as well. Volumetric EM reconstructions rather than area data would be more helpful here, if that capability is available to the authors.

Figure S7- (page 10) The differential gene expression data for the mitochondria were seem a bit stretched. The log₂FC values reported for differential expression were all <1.0, which suggests at best a minimal effect.

The Methods indicate that many of the study's endpoints were assessed at 10 weeks in the shiverer chimeras. But past studies using this model have used timepoints twice as long; this 10-week timepoint may be too short a time for oligodendrocyte differentiation, whether treated with metformin or not. Do the authors have longer transplant survivals by which to assess their metformin effects?

Minor points

The authors report significant changes in the mitochondrial, structure in the axons as well as the targeted OPCs and oligodendrocytes, and conclude that the metformin effect is not cell-specific. This is not surprising, but is it clear that these effects are direct drug effects? Metformin may change blood glucose, blood pressure, both systemic and cerebral blood flow, and a variety of other parameters, such that some of its effects as reported here might be secondary; this should at least be mentioned.

The authors' report much useful scRNA-Seq data as to the composition of OPCs in their different preparations. This work is of interest and stands on its own, but there are a number of protocols for producing OPCs and oligodendrocytes from pluripotent stem cells. Have the authors compared the scRNA expression patterns and composition of the cells produced by their differentiation protocol (Livesey et al), to those used by other labs? If not, they should at least mention the possibility that the results of metformin might vary depending upon how the OPCs that they are targeting were derived.

Similarly, the data in this paper were all derived from one human ESC line, RC17, but no genomic data are provided as to the genomic integrity of these cells, whether via CGH array or whole genome. Yet the metformin response in the differentiated OPCs and oligodendrocytes might be influenced by the mutational burden, if any, of the underlying ES line – a particular issue here as these cells were previously edited to express GFP, and the editing itself can be mutagenic. This should at least be discussed, but if genomic data for either the parental line or the OPCs are available, it would be valuable to add them.

While the authors description of the composition of their OPC cultures is detailed, it does raise some questions. In figure S3A the high expression of SOX2 is of particular concern; SOX2 is not expressed by oligodendroglia, but rather by much earlier neural stem cells before lineage restriction. The authors may wish to tighten up the cell type calls in their composition analysis.

Reviewer #2

(Remarks to the Author)

This review falls within the Nature Communications initiative to facilitate training in peer review for ECRs, which means it is a co-review, thus it will be written in the first-person plural.

The manuscript by Kazakou and colleagues aims to investigate whether metformin influences the ability of human OPCs to differentiate into oligodendrocytes and myelinate the axons. The ability of metformin to stimulate myelin repair has been previously validated in two rodent models of demyelination. In addition, it has been shown that this therapeutic agent "rejuvenates" aged rat OPCs and renders them responsive to pro-differentiation treatments. Importantly, several clinical trials to test the efficacy of metformin are ongoing in multiple sclerosis (MS) as well as other neurodegenerative diseases in which myelination is affected. Because functional differences between rodent and human oligodendroglial cells have been clearly demonstrated, validating the efficacy of metformin in human/human-like setting is of utmost importance for designing treatments to prevent progression in MS. The study uses 3 models to evaluate this effect, one of these being chimeric mice with human oligodendrocyte-mediated myelination, which is highly valuable to evaluate the interactions of human oligodendroglia with the axons and investigate the myelination potential of these cells under physiological conditions that cannot be reproduced in a dish. Once the specific candidate genes increased by metformin in oligodendrocytes are identified, the authors analyze whether these are also increased in oligodendroglia from MS patients treated with metformin. Our general evaluation is that this study is well performed and of great interest for the readers working on oligodendrocytes, myelination/remyelination, cell therapy, and regenerative therapies in MS in general, and that the results largely support the conclusions. However, we have noticed specific issues that we believe should be addressed before publication:

1. Page 4, Fig1: what is the composition of monolayer cultures with regard to oligodendroglial cells? The information on page 5 states scRNA seq data show that out of 19462 cells, 3369 (around 17%) are Olig2+, but what are the %s of O4+ and MBP+ cells, as determined by immunocytochemistry? As the authors quantified the differences between control and metformin-treated cells, they must have these data.

2. Related to previous point: the authors define the immature oligo population as Olig2+O4+ and the mature ones as O4+MBP+. However, in rodent cells, O4 expression is maintained in mature cells. In addition, some studies characterizing human cells report higher overlap between O4 and MBP than between O4 and PDGFRalpha. Do the authors mean that the immature cells were quantified as O4+MBP- ? Otherwise, the data presented in Fig.S1A would rather refer to both late progenitors and mature cells? Or was it previously confirmed that under these specific conditions O4 expression is downregulated with differentiation?

3. Page 5: "since our hESC-derived oligodendroglia were primarily patterned towards a dorsal character", do the authors mean "caudalized", as stated in the original publication (Livesey et al 2016), which would indeed justify the comparison with spinal cord scRNAseq dataset?
4. Page 6: "we instead integrated our hESC-derived oligodendroglia data with prenatal OPCs (3593) subsetted from a publicly available snRNAseq dataset from 106 donor brains of 13 to 40-week-old human fetuses (which do not yet contain oligodendrocytes) " - we believe the authors should revise this statement as human myelination begins in the third trimester of gestation, thus at this stage oligodendrocytes are present in the brain (Hasegawa et al., 1992; Jakovcsek et al., 2009 etc).
5. In human organoids, MBP fluorescence significantly increases in response to metformin treatment. Is this due to increased OPC proliferation and/or differentiation and/or oligo maturation, or more MBP production per cell? This could be answered by comparing Olig2+ (or PDGF α +) cell numbers, APC or NogoA+ cells numbers and potentially also the percentage of oligo lineage (Olig2+) cells expressing mature markers.
6. Page 8 and figure 3. This comment refers to the graph in Figure 3B, general characterization of rodent axons by human cells. Are the replicates included in this graph mice pooled from control and metformin-treated groups, or are these data from a separate experiment under control conditions? It would be useful to specify this in the figure legend and/or results text for clarity.
7. Chimeric mice were generated by transplanting the cells treated in the same way as those grown in monolayers except that differentiation induction was not performed. Thus, the transplanted cells were patterned in the same way as the monolayer cells. Monolayer cell data were compared to the spinal cord oligos RNAseq data, while transplanted cell data was compared to the human cortical dataset. We understand that this was done because the environment (corpus callosum) is an important determinant of transplanted cell behaviour. However, rodent experiments show that while different populations of OPCs can compensate for each other's functions to a certain extent, sometimes this compensation is not optimal suggesting that OPCs conserve their regional identity. What would be the results if transplanted cell data were compared to the spinal cord dataset, as the monolayer cultures were?
8. It has been previously shown that if healthy human cells are transplanted into shiverer neonates, they outcompete the endogenous MBP mutant oligodendroglia during myelination. Here the question is, because the quantification of axons includes only "myelinated" versus "unmyelinated" category and the authors state that only axons with compact myelin were considered, does this mean that there are no axons with uncompacted myelin (myelination by mouse cells) or that these axons are included in "unmyelinated" category?
9. Page 8: "finding a significant increase in the percentage of myelinated axons (and concomitant decrease in unmyelinated axons) after metformin treatment compared to vehicle control (6.76 mean fold change difference \pm 2.28 SEM) (Figure 3C)." When looking at the graph, the difference does not seem to be 6-7 fold, do the authors mean that the mean difference is of 6.76% ?
10. Were there differences in MBP fluorescence in the corpus callosum of chimeric mice treated with vehicle vs metformin?
11. Are there differences in HuNu+Olig2+ numbers and/or HuNu+APC(NogoA+) numbers? Or in the % of human oligodendroglia that is differentiated?
12. The data on mitochondria are beautiful, interesting, and convincing. Is the effect on axonal mitochondria limited to myelinated axons? Regarding this point, when one reads the data, the question arises on whether this effect in myelinated/remyelinated axons, concomitant to decreased G ratio might be a consequence of the changes in G ratios. However, we closely checked the Neumann 2019 reference, and this paper did not observe differences in G ratios in control vs metformin groups while differences in axonal mitochondria in these same tissues are reported in the current manuscript. We believe it may be useful to include this information in the discussion.
13. The authors mention 132 upregulated and 54 downregulated genes in chimeric mice human oligodendroglia and show some of these are also upregulated in mouse oligodendroglia. The categories they mention strongly suggest metabolic effects. Because CNS metabolism is an interplay between different cell types and the authors also have data on other cell types from vehicle vs metformin treated brains, it would be extremely interesting to analyse whether genes related to specific metabolic pathways are modulated in astrocytes and/or microglia, which could affect metabolic exchanges with oligodendrocytes (for example, via MCTs) and modulate myelination which may potentially increase our understanding on the mechanism by which metformin may be stimulating myelination, thus nicely complementing the discussion. Although many metabolic pathways are modulated at translational/post-translational level rather than that of gene expression, RNA seq data can sometimes provide important clues. These reviewers personally would be extremely interested in these data, although we understand that the authors may prefer to save the data for future manuscripts.
14. So, Ndufa11 upregulation in chimeras is human oligo-specific?
15. A couple of points related to discussion:
 - a. The authors mention that mouse studies show an effect of metformin on myelination by young and aged OPCs while in rats the effect is observed only in aged OPCs. However, to our knowledge, Neumann et al article did not test remyelination in young rats, they just observed no effect on in vitro young OPC differentiation.
 - b. Page 15 "However, our finding may also be explained by an increase in size or branching of mitochondria, which can occur via mitochondrial fusion, thought to contribute to enhancement of the mitochondrial networks within cells, improving energy supply and distribution"-changes in mitochondria could also reflect a switch in energy fuel utilization (glucose vs fatty acids, ketones etc).
 - c. Any thoughts on persistent expression of Sox2 by hES-derived oligos in all 3 models?
16. Please revise the following sentences:
 - a. Page 5 "To discover whether any of these cell types were similar to adult tissue types, and since our hESC-derived oligodendroglia were primarily patterned towards a dorsal character, we next integrated our data with our previously published snRNAseq adult human spinal cord dataset using canonical correlation analysis (CCA) followed by clustering with Seurat, since our hESC-derived oligodendroglia were primarily patterned towards a dorsal character."
 - b. Page 11: ""This is different from in our previous rat model"
 - c. In Supp. Fig.2 legend : "(A) The ANN was trained to recognize single-cell gene expression profiles of human spinal cord oligodendroglia using the union of variable highly variable features between human and hESC-derived monolayer

oligodendroglia.”

Reviewer #3

(Remarks to the Author)

In their manuscript, Katzakou et al. aimed to examine:

1. The similarity in RNA signatures among human ESC-derived oligodendrocytes in monocultures, organoids, and after transplantation into immunodeficient mice.
 2. The effect of metformin on oligodendroglial lineage cells within the transplantation model.
 3. Whether oligodendrocytes from MS brain donors treated with metformin exhibited similar mRNA signatures.
- The authors state that metformin enhanced the expression of myelin proteins and/or formation of myelin sheaths across all models. Additionally, metformin increased mitochondrial area in both transplanted human cells and mouse axons, along with upregulation of transcripts related to mitochondrial function and metabolism.
- While this is, in principle, an interesting study, several methodological issues limit my enthusiasm for its findings.

General comments:

1. The manuscript does not conclusively demonstrate that metformin enhances oligodendroglial function via alterations in mitochondrial metabolism, as suggested in the title. The authors show that transplantation of their spheroids results in more mature ESC-derived oligodendrocytes and that metformin increases MBP densities in spheroids. However, evidence is lacking for increased MBP+ cell numbers in vivo or enhanced myelination by human oligodendrocytes. Additionally, while metformin alters the expression of mitochondrial genes, there are no experiments linking these changes to functional modifications in mitochondrial metabolism or oligodendroglial function.
2. This manuscript covers two different topics, on the one hand comparison of different approaches for the differentiation of ESC-derived oligodendrocytes, on the other hand the investigation of metformin-mediated effects on these cells. It seems, at least to this reviewer, that the authors could not really decide, on which topic they want to focus, which may contribute to lack of in-depth analyses.
3. Please briefly discuss ongoing clinical trials in this field. Given the number of clinical studies with different patient cohorts, the authors should explain why further preclinical studies are necessary.
4. The Materials and Methods section indicates that all experiments were performed using only one stem cell line. This does not align with current standards in human stem cell research. The use of multiple lines is recommended to account for the well-known variability between lines. Why was a human embryonic stem cell line chosen instead of the widely available iPSCs from healthy and MS donors?
5. The study heavily relies on scRNAseq, which this reviewer considers a limitation. While scRNAseq facilitates comparisons with prior studies, the correlation between RNA and protein data is limited. Additional protein or functional analyses would strengthen the study.
6. Quantitative data on oligodendroglial markers and myelination at the protein level are lacking. For instance, what percentages of OLIG2+, O4+, and MBP+ cells are observed in monocultures and spheroids? What is the extent of myelination in the spheroid models in vitro?
7. The authors do not provide direct evidence that metformin promotes myelination by transplanted human cells.

Specific comments:

1. The rationale behind the treatment schemes for monocultures (7 days) and organoids (days 60–70) is unclear. Please clarify the exact timing of treatments and whether the chosen metformin concentration (100 μ M) is achievable in the human CNS. Were dose-response curves or alternative incubation periods tested?
2. Figure 1D and E: Does the distribution of oligodendroglial subsets based on RNA align with protein-level findings?
4. Figure 2B: Scale bars are missing. Please include vehicle controls for comparison.
5. Figure 2C: MBP density might indicate changes in either the number of MBP+ cells or the size/branching of individual cells. Could the authors provide data on the proportion of MBP+ cells among total cells (or SOX10+ oligodendroglial cells)?
6. Figures 2D and E: Only 301 of ~12,000 cells belong to the oligodendroglial lineage, with a small fraction in the mature Oligo2 subcluster. Are these findings consistent with protein data? If not, this discrepancy may highlight limitations of the scRNAseq approach. If yes, this may suggest that the spheroid model requires further optimization to enhance oligodendroglial differentiation.
7. Figure 2G: The Oligo2 subcluster shows general similarity to various in vivo subclusters but no strong alignment with any specific one. Could the authors discuss whether this result was expected and what does this finding mean?
8. Figure 3B: Please provide comparisons to non-transplanted controls and include double-staining for huNu and MBP. If each dot represents an animal, specify this in the figure legend.
9. Figure 3C: How do the authors attribute increased myelinated axons specifically to human cells? Since shiverer mice exhibit loose myelin wrapping, can the authors quantify MBP+ oligodendrocytes or axons with compact myelin in transplanted mice treated with vehicle versus metformin? Is there any change in MBP/NF colocalization in metformin treated animals?
10. Figures 4A and C: How do the authors differentiate between myelin sheaths generated by mouse versus human oligodendrocytes?
11. Figures 4B and D: The inclusion of pseudo-replicates inflates sample numbers. Please use averages for mitochondrial or myelinated axon area per mouse and compare vehicle-treated versus metformin-treated animals (n=4 and 5, respectively).
12. Figure 4: While metformin upregulates certain genes (e.g., EIF1, NDUFA11, COX8A), this upregulation is not cell-type-specific. Experiments linking these changes to oligodendroglial differentiation or myelination are lacking, limiting the conclusions drawn.
13. Figure 4: Data from MS tissue samples have low statistical power due to small sample sizes; therefore, the relevance of

the findings remains unclear.

14. Discussion: The authors should address the limitations of their models. For instance, stem cell-derived oligodendrocytes represent “young” cells lacking aging signatures and likely behave differently from adult human OPCs or mature oligodendrocytes (as shown in rodents). Moreover, the models do not replicate inflammatory environments characteristic of MS.

Reviewer #4

(Remarks to the Author)

Version 1:

Reviewer comments:

Reviewer #1

(Remarks to the Author)

The authors have addressed my comments and requests very well. The paper's very solid now, and adds to the preclinical data package suggesting the therapeutic value of metformin in optimizing the state of central myelin and potentiating white matter health. I've no further requests for revision, though I would encourage the authors to advocate institutionally that shiverers be allowed to live beyond 70 days in the UK; the real experimental value of these chimeras is not realized until preterminally, and this experiment in particular would have likely yielded far more compelling quantitative data had longer survival points been allowable.

Reviewer #2

(Remarks to the Author)

This review falls within the Nature Communications initiative to facilitate training in peer review for ECRs, which means it is a co-review, and thus it is written in plural.

In the revised version of the manuscript, the authors have entirely addressed our comments on the previously submitted version of their manuscript. They now show convincingly that metformin increases myelination by human cells, and that it alters mitochondria in all cell types of the CNS. These findings are of great relevance for developing treatments for MS and other neurodegenerative diseases. We therefore strongly recommend accepting this article for publication. However, we also suggest to closely check the authors' affiliations prior to publication, as we suspect there might be a mismatch for some authors.

Reviewer #3

(Remarks to the Author)

The authors have addressed all my concerns.

Reviewer #4

(Remarks to the Author)

RESPONSE TO REVIEWER COMMENTS

Reviewer #1

The paper is solid and interesting, a logical extension of the groups' prior studies, and an important preclinical study if the intent is to proceed to clinical trial of this strategy. I do have some concerns though, that the paper would benefit from addressing.

Major points:

Figure 1-The author report that a week (7d) of metformin increased the incidence of both O4+ and MBP+ oligodendrocyte stages. Yet the 0.7-fold change they report is underwhelming, and the data are only reported as fold-change; we have no idea what the absolute number and relative percentages are, without which fold-change data are difficult to interpret. In that regard, figure 1B seems to indicate that the MBP percentages are quite low – not surprising, since human OPCs express little MBP in OPC monolayer culture - typically only with strong T3/IGF1 stimulation unless axons are present - and even then not until longer time points, since human oligodendrocytes take longer to mature in most culture protocols. Overall, these data might have looked better with longer treatment times. It would be helpful if the authors have such data to report, along with the quantitative data – numbers and percentages – upon which their conclusions are based.

We agree that it is not very surprising that the number of human cells expressing MBP are low as they are immature, at least in part due to being rather fetal-like. When maintaining our monolayer cultures for extended periods, instead of an increase in MBP⁺ cells, we consistently observe a decline in the percentage of OLIG2⁺ cells and an overgrowth of astrocytes (both by IHC and in our scRNAseq data), dominating the cultures. Culture survival is also limited over extended culturing periods (see below).

i) immunofluorescence staining showing reduction of OLIG2⁺ cells (red) over time (Hoechst=all nuclei, blue). ii) quantification of OLIG2⁺ over time (ICC) n=number of separate cultures, mean±SEM. iii) scRNAseq quantification of percentage of OLIG2⁺ (RNA) cells in monolayer cultures over time iv) scRNAseq quantification of percentage of GFAP⁺ (RNA) cells in monocultures over time.

We reported fold-change in O4⁺ and MBP⁺ cells as the cultures are variable between batches, but we have now also added the graphs reporting percentages of OLIG2⁺ cells that are O4⁺ or MBP⁺ to supplementary data **Figure 1A** - reproduced below.

A) Fold change (FC) difference or percentages of OLIG2⁺O4⁺ or OLIG2⁺MBP⁺ oligodendrocytes after treatment with metformin or clemastine compared to their respective vehicle-treated controls (ddH₂O or DMSO). n = 5 (O4) or n=6 (MBP) differentiations with 4 technical repeats for each. Kolmogorov-Smirnov normality test with Dallal-Wilkinson-Lillie for p value, two-tailed unpaired t-test. Error bars SEM.

Figure 2C- The authors report that in organoids treated for 10 days, that MBP density was significantly increased. But “MBP density” is really qualitative, neither well-described nor strictly quantitative, and again, a fold-increase is reported rather than clear percentages. Also, a correlation with the detailed genomic data, relating the MBP data to the compositional data obtained in the scRNA-Seq analysis (figs. 2D and S2), would be informative.

We have now added a clearer description of how we performed the MBP quantification for the spheroids to the methods section as below:

“Automated analysis of MBP quantification was performed on 10micron-thick cryostat-sectioned spheroids. Sections were immunofluorescently labelled with an antibody against MBP and counterstained with Hoechst. Tile-Scan images of the whole section were acquired with the 10x objective using a Leica SP8 confocal microscope. A custom Fiji/ImageJ macro was used for image quantification. The total tissue section area was defined based on Hoechst staining. MBP signal was identified by immunofluorescence and reported as the percentage of MBP-positive area relative to the total tissue section area (termed MBP immunofluorescence area).”

We also now express the MBP-immunofluorescence results after metformin treatment in the spheroid cultures as the percentage of area of signal/total area rather than fold change for clarity (**Figure 2C** and below). Furthermore, we have added analysis of the percentage of mature CC1⁺ oligodendrocytes in the spheroid cultures, and those oligodendrocytes which are most mature, expressing the myelin protein MBP. The results of both show that metformin increases the amount of myelin but not the number of mature oligodendrocytes in the spheroid cultures (**Supplementary figure 4 A, B** and reproduced below).

We are unable to directly compare immunofluorescence measures after metformin with scRNAseq as we did not perform scRNAseq on the spheroids treated with metformin, just on the untreated spheroids. This was as our goal was to first identify the optimal human oligodendrocyte system for testing the effect of metformin on myelin or myelin surrogate measures.

Figure 3 (page 8)-Again the use of fold-change endpoints seems a bit confusing – the authors report that 16.15% of rodent callosal axons are myelinated in untreated chimeras and that this number increased by 6.76-fold after metformin-treatment – so does that mean that >100% of axon were myelinated? Yet Fig 3C shows a barely perceptible bump in the median percentage of myelinated axons to metformin. I may be misunderstanding the author’s argument here, but regardless, this section needs to be clarified, and again, actual numbers and percentages provided.

Many apologies, this reviewer is correct and this was a mistake. This was a percentage increase. However, we have changed all of the values in this section to numbers, as highlighted in the revised paper and reproduced below:

“We treated these chimeric animals with metformin daily by oral gavage for 21 days, between days 42-63 post transplantation and assessed compact myelination at 70 days post transplantation by electron microscopy, finding a significant increase in the percentage of myelinated axons (and concomitant decrease in unmyelinated axons) after metformin treatment compared to vehicle control (from 21.44 ± 2.3 SEM to 28.21 ± 1.9 SEM) (**Figure 3C**). We also observed a significant decrease in the average g-ratio between treated and control groups (from 0.84 ± 0.004 SEM to 0.81 ± 0.009 SEM), independent of axonal diameters, suggesting that treatment with metformin also increased myelin thickness (**Figure S5 G, H**).”

Figure 4 (page 9) The authors report that both glial and neuronal mitochondrial profiles increased to metformin, by 0.05-0.1-fold depending on size normalization. Yet despite the significance claimed in Fig. 4B, the sample size here seems far too low for a comparison made in 2D TEMs that is significantly confounded by slice orientation. The authors need to make clear how many cells were sampled in each of the few mice in each group, how many mitochondria/cell, and areas thereof; the fold change endpoint and relatively sparse data shown in 4C are not convincing. The statistics applied need to be better described here as well. Volumetric EM

reconstructions rather than area data would be more helpful here, if that capability is available to the authors.

Fortunately, the axons of the corpus callosum are orientated in a standard pattern, with fibres passing from anterior to posterior located in the centre of the corpus callosum whereas those passing laterally are at the top and bottom. This means that we can choose similar areas in each animal to have the fibres in the same direction and take sections so that most axons have transverse profiles, to allow standardisation for both the myelin thickness calculations and aiding the measurement of mitochondrial size.

We clarify numbers, we have also now moved our data for the mitochondria area as an average per axon/glia per animal from the supplementary data to the main **figure 4**, to lie next to the graphs of every value per individual axon/glia for completeness. We reproduce these below as well:

Figure 4B and D – showing both average mitochondria area per axon counted per mouse and per axon, as well as average mitochondria area per glial cell area counted per mouse and per glial cell area.

We have also improved the description of how we analysed the TEM images in the methods and copied below.

“For mitochondrial area quantification, 3× fields of 5.3 x 5.3 microns containing ~30–35 transverse profiles of axons in total per mouse (n = 5 metformin- and n = 4 vehicle-treated animals) and 7× fields 10.8 x 10.8 microns containing ~1–2 glia per mouse (n = 6 metformin- and n = 3 vehicle-treated animals) were traced manually and the size calculated using the Fiji/ImageJ TrackEM plugin. Values were analyzed both as averages per mouse and as individual measurements per axon/glia. Normality was assessed using the Kolmogorov-Smirnov test with the Dallal-Wilkinson-Lillie method for p-value calculation. Based on normality, statistical comparisons were performed using either a two-tailed unpaired t-test or the Mann-Whitney test.”

We agree that it would be interesting to do volumetric EM reconstructions in the future to better analyse mitochondria shape differences.

Figure S7- (page 10) The differential gene expression data for the mitochondria were seem a bit stretched. The log2FC values reported for differential expression were all <1.0, which suggests at best a minimal effect.

We used this cut off as a log2 fold change (log2FC) of 0.5 corresponds to an approximate 41.4% increase in expression levels, and this is the threshold suggested in the benchmarking paper published in 2023 “Benchmarking integration of single-cell differential expression | Nature

Communications". This is a stricter threshold than the standard default setting of 0.25 in the Seurat package. While setting a higher log2FC threshold can reduce the number of identified differentially expressed genes to those with more pronounced changes, it may also exclude genes with smaller yet potentially biologically relevant differences. Our confidence in this also is increased by expressed genes being in similar pathways and our validations. For example, TOMM20 (translocase of outer mitochondrial membrane-20) and CHCHD2 are both implicated in mitochondrial biology pathways and are also upregulated after treatment with metformin (also labelled on volcano plot). TOMM20 is located on the outer mitochondrial membrane and considered as a marker of mitochondrial metabolic activity. CHCHD2 is a transcription factor that regulates mitochondrial dynamics and promotes oxidative phosphorylation. We have added these comments about these gene transcripts to the results section to add confidence.

The Methods indicate that many of the study's endpoints were assessed at 10 weeks in the shiverer chimeras. But past studies using this model have used timepoints twice as long; this 10-week timepoint may be too short a time for oligodendrocyte differentiation, whether treated with metformin or not. Do the authors have longer transplant survivals by which to assess their metformin effects?

In the UK, the animal regulation rules (from the UK Home office) state that we are only allowed to maintain *Shiverer:Rag2^{-/-}* mice until day 70 for animal welfare reasons. This is different from other countries, and different from what was originally set up by the Steve Goldman Lab in the USA. We agree it would be interesting to look at later timepoints, but fortunately we are easily able to see myelination from our human cells at this timepoint in our animals.

Minor points

The authors report significant changes in the mitochondrial, structure in the axons as well as the targeted OPCs and oligodendrocytes, and conclude that the metformin effect is not cell-specific. This is not surprising, but is it clear that these effects are direct drug effects? Metformin may change blood glucose, blood pressure, both systemic and cerebral flood flow, and a variety of other parameters, such that some of its effects as reported here might be secondary; this should at least be mentioned.

We agree that perhaps the effect of metformin on other cells besides oligodendroglia is not too surprising, however it has not yet been reported in the brain. Thank you for also pointing out other potentially relevant effects and we have added these as possible influences to the discussion section of the article.

The authors' report much useful scRNA-Seq data as to the composition of OPCs in their different preparations. This work is of interest and stands on it own, but there are a number of protocols for producing OPCs and oligodendrocytes from pluripotent stem cells. Have the authors compared the scRNA expression patterns and composition of the cells produced by their differentiation protocol (Livesey et al), to those used by other labs? If not, they should at least mention the possibility that the results of metformin might vary depending upon how the OPCs that they are targeting were derived.

We indeed were able to do this, choosing two other datasets of differentiation of hES cells towards oligodendroglial fate using retinoic acid GSE223599 and GSE146373^{1,2}.

Integration of these datasets showed that the main OPC and oligodendrocyte clusters were present in all of these datasets, with some differences in the proportion of smaller clusters as shown below. These additional datasets did not add metformin (but did have reporters/other manipulations) (see below).

UMAP of integrated datasets, feature plot showing expression of MBP (mature oligodendrocytes) and PDGFRA (OPC) transcript markers, and separated UMAPs showing each additional dataset and our dataset.

Similarly, the data in this paper were all derived from one human ESC line, RC17, but no genomic data are provided as to the genomic integrity of these cells, whether via CGH array or whole genome. Yet the metformin response in the differentiated OPCs and oligodendrocytes might be influenced by the mutational burden, if any, of the underlying ES line – a particular issue here as these cells were previously edited to express GFP, and the editing itself can be mutagenic. This should at least be discussed, but if genomic data for either the parental line or the OPCs are available, it would be valuable to add them.

To ensure the genomic integrity of our cell-line we indeed had performed a SNP array using the CytoSNP 850 K BeadChip from Illumina and analysed in GenomeStudio 2.0 with the plug-in cnvPartition 3.2.0. compared to the RC17 reference line. This analysis showed that there were no karyotype abnormalities and we report the CNVs comparing to the reference genome in (**Supplementary Data - CNVs.xlsx**). We have added this to the methods section and supplementary data.

While the authors description of the composition of their OPC cultures is detailed, it does raise some questions. In figure S3A the high expression of SOX2 is of particular concern; SOX2 is not expressed by oligodendroglia, but rather by much earlier neural stem cells before lineage restriction. The authors may wish to tighten up the cell type calls in their composition analysis.

We agree that the high expression of SOX2 is different from what we expected, which is one

reason for our reporting of it. However, we do not believe that this expression means that these cells are all NSCs as they also express other markers of mature oligodendroglia such as for example MBP – a marker of mature oligodendrocytes. Furthermore, this is present at the protein level not just at the RNA level. Instead of seeing this as a misclassification of cells, we see this more as evidence that these cells are fundamentally immature as they have not switched off their stem/progenitor transcripts. We even wonder whether this continued expression of SOX2 is a survival advantage for these cells in culture and at least part of the reason why human ES-derived OPCs do not myelinate well/at all in 2-dimensional cultures, but this is speculation. We also think it is of interest that the expression levels of SOX2 from the scRNAseq are highest in monocultures and lowest in oligodendrocytes from the chimera, perhaps also reflecting their increase in maturity. We have added the immunohistochemistry pictures below to **supplementary figure 3B** to illustrate the co-expression of these protein markers. Red arrowheads indicate OLIG2⁺SOX2⁺ cells, while white arrowheads indicate OLIG2⁺SOX2⁻ cells.

Reviewer #2 (Remarks to the Author) combined with Reviewer 4:

Our general evaluation is that this study is well performed and of great interest for the readers working on oligodendrocytes, myelination/remyelination, cell therapy, and regenerative therapies in MS in general, and that the results largely support the conclusions. However, we have noticed specific issues that we believe should be addressed before publication:

1. Page 4, Fig1: what is the composition of monolayer cultures with regard to oligodendroglial cells? The information on page 5 states scRNA seq data show that out of 19462 cells, 3369 (around 17%) are Olig2+, but what are the %s of O4+ and MBP+ cells, as determined by immunocytochemistry? As the authors quantified the differences between control and metformin-treated cells, they must have these data.

Yes, we do indeed have these data as shown below. As determined by immunocytochemistry, the mean percentage of OLIG2⁺O4⁺ cells is 13.82 ± 1.2 SEM and of OLIG2⁺MBP⁺ 24.74 ± 4.2 SEM in monolayers after 1 week of maturation.

2. Related to previous point: the authors define the immature oligo population as Olig2+O4+ and the mature ones as O4+MBP+. However, in rodent cells, O4 expression is maintained in mature cells. In addition, some studies characterizing human cells report higher overlap between O4 and MBP than between O4 and PDGFRA. Do the authors mean that the immature cells were quantified as O4+MBP- ? Otherwise, the data presented in Fig.S1A would rather refer to both late progenitors and mature cells? Or was it previously confirmed that under these specific conditions O4 expression is downregulated with differentiation?

We agree that there is an overlap in markers between immature oligodendroglia and more mature, with the pattern of expression starting at markers such as PDGFRA, moving to O4 in the intermediate maturity cells and then to MBP (or the other myelin protein markers) in the more mature oligodendrocytes. This is probably best illustrated in the cartoon below.

We were interested in how metformin increases differentiation from the OPCs into O4+ and then MBP+ ones. We considered the drug effect on that continuum using the single markers, shifting cells to the right (more mature) with the drugs, which we see as positive. To make this clearer, we have been more careful with our language, calling the O4+/MBP+ cells “more mature” than the “intermediate” O4+/MBP- ones.

3. Page 5: “since our hESC-derived oligodendroglia were primarily patterned towards a dorsal character”, do the authors mean “caudalized”, as stated in the original publication (Livesey et al 2016), which would indeed justify the comparison with spinal cord scRNAseq dataset?

Thank you, that is exactly what we mean and have corrected our wording.

4. Page 6: “we instead integrated our hESC-derived oligodendroglia data with prenatal OPCs (3593) subsetted from a publicly available snRNAseq dataset from 106 donor brains of 13 to 40-week-old human fetuses (which do not yet contain oligodendrocytes) “- we believe the authors should revise this statement as human myelination begins in the third trimestre of gestation, thus at this stage oligodendrocytes are present in the brain (Hasegawa et al., 1992; Jakovceski et al., 2009 etc).

You are indeed correct, and we have revised this to “no oligodendrocytes in the first two trimesters but some in the third trimester.”

5. In human organoids, MBP fluorescence significantly increases in response to metformin treatment. Is this due to increased OPC proliferation and/or differentiation and/or oligo maturation, or more MBP production per cell? This could be answered by comparing Olig2+ (or PDGFalpha +) cell numbers, APC or NogoA+ cells numbers and potentially also the percentage of oligo lineage (Olig2+) cells expressing mature markers.

We have addressed this question (which was similar to that of Reviewer 1) using immunohistochemistry using antibodies to CC1 and MBP to quantify the number of cell bodies and comparing this to the MBP area signal (relates to processes and myelin sheaths). This shows that metformin treatment of our human organoids does not increase the number of mature CC1⁺ or MBP⁺ oligodendrocytes (a minority of more mature MBP⁺ oligodendrocytes can be CC1⁺, as shown by these data). However, metformin treatment does increase MBP area, suggesting an increase in myelin production per cell rather than an increase in the number of cells. These data are added to **Supplementary Figure 4A, B**.

New Figure 2C

Supp Figure 4A & B

6. Page 8 and figure 3. This comment refers to the graph in Figure 3B, general characterization of rodent axons by human cells. Are the replicates included in this graph mice pooled from control and metformin-treated groups, or are these data from a separate experiment under control conditions? It would be useful to specify this in the figure legend and/or results text for clarity.

Thank you, this was absent from the figure legend and we have now corrected this. These are from control mice with human cells but without treatments (not pooled with treated animals).

7. Chimeric mice were generated by transplanting the cells treated in the same way as those grown in monolayers except that differentiation induction was not performed. Thus, the

transplanted cells were patterned in the same way as the monolayer cells. Monolayer cell data were compared to the spinal cord oligos RNAseq data, while transplanted cell data was compared to the human cortical dataset. We understand that this was done because the environment (corpus callosum) is an important determinant of transplanted cell behaviour. However, rodent experiments show that while different populations of OPCs can compensate for each other's functions to a certain extent, sometimes this compensation is not optimal suggesting that OPCs conserve their regional identity. What would be the results if transplanted cell data were compared to the spinal cord dataset, as the monolayer cultures were?

We indeed did compare monolayer cells to our human spinal cord dataset and organoid cells to our human cortical white matter dataset for the reasons this reviewer has stated. However, with the transplanted cells into the shiverer mice, we compared the cells to the combined spinal cord and brain dataset and we have made this clearer in the legend. However, the point still stands, and we have redone the CCA for these cells against each region separately, as shown below. The UMAPs are similar, with most overlap with OPCs but some overlap with oligodendrocytes. We have added this to **new supplementary figure 6**.

8. It has been previously shown that if healthy human cells are transplanted into shiverer neonates, they outcompete the endogenous MBP mutant oligodendroglia during myelination. Here the question is, because the quantification of axons includes only “myelinated” versus “unmyelinated” category and the authors state that only axons with compact myelin were considered, does this mean that there are no axons with uncompacted myelin (myelination by mouse cells) or that these axons are included in “unmyelinated” category?

In *Shiverer* mice lacking MBP, there is no compact myelin, which is of course their advantage in these experiments. Wraps of uncompacted membrane (from mouse cells) do occur in *Shiverer* mice without human cells transplanted in as shown in the electron microscopy pictures below (scale bar 0.5um).

We included these as unmyelinated axons previously, but we have now been clearer about explaining this in the text:

“Myelinated axons are defined here as those with compact myelin (of human origin) and axons surrounded by uncompacted membranes (likely mouse origin) are grouped with unmyelinated axons.”

9. Page 8: “finding a significant increase in the percentage of myelinated axons (and concomitant decrease in unmyelinated axons) after metformin treatment compared to vehicle control (6.76 mean fold change difference \pm 2.28 SEM) (Figure 3C).” When looking at the graph, the difference does not seem to be 6-7 fold, do the authors mean that the mean difference is of 6.76% ?

We apologise – two reviewers spotted this and are correct – thank you! We have now changed this.

10. Were there differences in MBP fluorescence in the corpus callosum of chimeric mice treated with vehicle vs metformin?

When we measured the area of MBP staining on chimeric tissue over the corpus callosum between the control and metformin-treated mice, we did not see a statistically significant difference (though the trend is an increase). However, this was detectable by electron microscopy.

11. Are there differences in HuNu+Olig2+ numbers and/or HuNu+APC(NogoA+) numbers? Or in the % of human oligodendroglia that is differentiated?

There are not differences in the percentage of all cells which are human oligodendroglia (HuNu⁺Olig2⁺) between control and metformin groups nor in the percentage of these cells which have differentiated into CC1⁺ oligodendrocytes (an alternative marker of mature oligodendrocytes compatible with our HuNu antibody). This is consistent with our data from the organoid cultures, where again we found no change in numbers of mature oligodendrocytes (again as measured by CC1⁺ and by MBP⁺) but an increase in MBP⁺ signal attributable to myelin sheaths. We have added these graphs to **supplementary Figure 6 (E, F)** and a comment in the results section.

Supplementary Figure 6:

(E) Percentage of all cells which are OLIG2+HuNu+CC1+ oligodendrocytes, and (F) of all human oligodendroglia which are CC1⁺ in the chimeric mouse corpus callosum 70 days post transplantation comparing control and metformin-treated groups. Points are individual mice, mean ± SEM, p values stated.

12. The data on mitochondria are beautiful, interesting, and convincing. Is the effect on axonal mitochondria limited to myelinated axons?

Thank you, we were excited by the mitochondrial changes as well.

Although metformin treatment enhances the overall amount of myelin, it also increases mitochondrial area regardless of whether axons are myelinated or not. Thus, it is interesting that its effect on mitochondrial area is independent of the myelination status. We have added this to

supplementary figure 7A (and reproduced below) and added a comment in the results and discussion.

New Supplementary Figure 7A

Regarding this point, when one reads the data, the question arises on whether this effect in myelinated/remyelinated axons, concomitant to decreased G ratio might be a consequence of the changes in G ratios. However, we closely checked the Neumann 2019 reference, and this paper did not observe differences in G ratios in control vs metformin groups while differences in axonal mitochondria in these same tissues are reported in the current manuscript. We believe it may be useful to include this information in the discussion.

Thank you for this careful reading. We agree, and this is consistent with it being an effect independent of compact myelination. We have added information to the results and discussion to highlight this.

13. The authors mention 132 upregulated and 54 downregulated genes in chimeric mice human oligodendroglia and show some of these are also upregulated in mouse oligodendroglia. The categories they mention strongly suggest metabolic effects. Because CNS metabolism is an interplay between **different cell types** and the authors also have data on other cell types from vehicle vs metformin treated brains, it would be extremely interesting to analyse whether **genes related to specific metabolic pathways are modulated in astrocytes and/or microglia**, which could affect metabolic exchanges with oligodendrocytes (for example, via MCTs) and modulate myelination which may potentially increase our understanding on the mechanism by which metformin may be stimulating myelination, thus nicely complementing the discussion. Although many metabolic pathways are modulated at translational/post-translational level rather than that of gene expression, RNA seq data can sometimes provide important clues. These reviewers personally would be extremely interested in these data, although we understand that the authors may prefer to save the data for future manuscripts.

We are also very interested in this. We showed a limited part of this in the supplementary data figures for the 3 genes that we validated, but we decided that the best way to show this was to include the volcano plots of differentially expressed genes per cell type between control and metformin-treated mouse cells in **supplementary figure 9B** (reproduced below) (and data tables S2-S5 as before).

These analyses do indeed reveal interesting evidence of metformin’s effect on metabolic pathways, especially if we relax the log fold change (LFC) threshold to >0.25. For example, *ChChd2* expression is not only increased in human oligodendroglia, but also increased in mouse oligodendrocytes (LFC>0.5), and if we use a LFC threshold of >0.25, also in neurons, astrocytes and microglia. *Timm10b* expression is an inner mitochondrial membrane protein which mediates import and insertion of transmembrane protein complexes into the inner membrane, and this is increased in mouse oligodendrocytes and microglia (LFC>0.25). We have added an additional comment in the results section reflecting this, and the reviewers are correct that we continue to be interested in the lab in mitochondrial dynamics after metformin use.

14. So, *Ndufa11* upregulation in chimeras is human oligo-specific?

In our white matter samples from this model, this is correct. We did not detect an upregulation of *Ndufa11* in the mouse cells – either in the mouse oligodendrocytes alone or in all mouse cells (figure below). We have made sure this is clear in the text.

Expression of *Ndufa11* in mouse oligodendroglia

Expression of *Ndufa11* in all mouse cells

15. A couple of points related to discussion:

a. The authors mention that mouse studies show an effect of metformin on myelination by young and aged OPCs while in rats the effect is observed only in aged OPCs. However, to our knowledge, Neumann et al article did not test remyelination in young rats, they just observed no effect on in vitro young OPC differentiation.

This is correct and we have altered this accordingly.

b. Page 15 “However, our finding may also be explained by an increase in size or branching of mitochondria, which can occur via mitochondrial fusion, thought to contribute to enhancement of the mitochondrial networks within cells, improving energy supply and distribution”-changes in mitochondria could also reflect a switch in energy fuel utilization (glucose vs fatty acids, ketones etc).

Yes, we agree, and we have added a sentence to this effect.

c. Any thoughts on persistent expression of Sox2 by hES-derived oligos in all 3 models?

We also did not expect the high expression of SOX2 in all three models, which is one reason for our reporting of it. We see this as evidence that these cells are still fundamentally immature as they have not switched off their stem/progenitor transcripts. We even wonder whether this continued expression of SOX2 is a survival advantage for these cells in culture and at least part of the reason why human ESC-derived OPCs do not myelinate well/at all in 2-dimensional cultures, but this is speculation. We also think it is of interest that the expression levels of SOX2 from the scRNAseq are highest in monocultures and lowest in oligodendrocytes from the chimera, perhaps also reflecting their increase in maturity in the chimeras. We have added the immunohistochemistry pictures below to **supplementary figure 3B** to illustrate the co-expression of these protein markers. Red arrowheads indicate OLIG2⁺SOX2⁺ cells, while white arrowheads indicate OLIG2⁺SOX2⁻ cells.

16. Please revise the following sentences:

a. Page 5 “To discover whether any of these cell types were similar to adult tissue types, and since our hESC-derived oligodendroglia were primarily patterned towards a dorsal character, we next integrated our data with our previously published snRNAseq adult human spinal cord dataset using canonical correlation analysis (CCA) followed by clustering with Seurat, since our hESC-derived oligodendroglia were primarily patterned towards a dorsal character.”

Corrected – thank you.

b. Page 11: “This is different from in our previous rat model”

Revised – thank you

c. In Supp. Fig.2 legend : “(A) The ANN was trained to recognize single-cell gene expression profiles of human spinal cord oligodendroglia using the union of variable highly variable features between human and hESC-derived monolayer oligodendroglia.”

Corrected – thank you.

Reviewer 3

General comments:

1. The manuscript does not conclusively demonstrate that metformin enhances oligodendroglial function via alterations in mitochondrial metabolism, as suggested in the title. The authors show that transplantation of their spheroids results in more mature ESC-derived oligodendrocytes and that metformin increases MBP densities in spheroids. However, evidence is lacking for increased MBP+ cell numbers in vivo or enhanced myelination by human oligodendrocytes. Additionally, while metformin alters the expression of mitochondrial genes, there are no experiments linking these changes to functional modifications in mitochondrial metabolism or oligodendroglial function.

We agree that we have shown that metformin alters mitochondrial structure and gene expression related to mitochondria in oligodendroglia (and other cells) and that metformin enhances remyelination, but not that the mitochondrial changes directly cause the improved remyelination. We have changed the title to reflect this.

We have now added evidence that metformin does not change MBP+ cell number in spheroids or chimeras (**Supplementary Figure 4A, B, supplementary figure 5E, F**), but that myelin sheaths are increased. We also show that Shiverer mice treated with metformin without a human oligodendroglial cell transplant do not produce compact myelin (**supplementary figure 7B**), showing that the human cells are necessary to do this.

These experiments are detailed further in our response to the specific comments below.

2. This manuscript covers two different topics, on the one hand comparison of different approaches for the differentiation of ESC-derived oligodendrocytes, on the other hand the investigation of metformin-mediated effects on these cells. It seems, at least to this reviewer, that the authors could not really decide, on which topic they want to focus, which may contribute to lack of in-depth analyses.

We smiled at this comment as indeed we did consider whether to split the paper into two – one for the comparison of the techniques and the other for the action of metformin. However, we feel that the research story works best with both combined, as we have written it and would prefer to keep it this way.

3. Please briefly discuss ongoing clinical trials in this field. Given the number of clinical studies with different patient cohorts, the authors should explain why further preclinical studies are necessary.

We have increased the discussion of these trials in the introduction. There are indeed several clinical studies currently on-going. However, none has yet reported their results in MS and we still do not know the mechanism of action of metformin in human brain cells. This study increases understanding of this mechanism, which may be important for design of future drugs which may be potentially more effective if the target is known, with fewer unwanted side effects.

4. The Materials and Methods section indicates that all experiments were performed using only one stem cell line. This does not align with current standards in human stem cell research. The use of multiple lines is recommended to account for the well-known variability between lines. Why was a human embryonic stem cell line chosen instead of the widely available iPSCs from healthy and MS donors?

Indeed, we chose to use one ES cell line – the RC17 human line. We specifically chose not to use iPSC cells from MS and controls for a variety of reasons. Firstly, as MS is not a single gene disorder, we cannot use isogenic controls. Secondly, our question was whether metformin improved myelin formation from human OPCs generally, rather than whether this would be different in oligodendrocytes from people with MS. Finally, we did not carry this out in multiple ES cell lines as we were using this technique to discover biology, which we then tried to validate using brain samples from human brains, as, in our opinion, this is a better way of gaining robust relevance to humans.

5. The study heavily relies on scRNAseq, which this reviewer considers a limitation. While scRNAseq facilitates comparisons with prior studies, the correlation between RNA and protein data is limited. Additional protein or functional analyses would strengthen the study.

We have added protein correlations for differentiation of monoculture oligodendroglia (**supplementary figure 1A**), of organoids (**supplementary figure 4A,B**) and of chimeras (**supplementary figure 5E,F**) to strengthen the paper.

We also now include western blots of hES-derived oligodendroglial cultures treated with metformin, for the mitochondrial proteins TOMM20 and CHCHD2 which are increased compared to the controls. TOMM20 is located on the outer mitochondrial membrane and considered as a marker of mitochondrial metabolic activity (<https://doi.org/10.1016/j.bbadis.2020.165962>) and CHCHD2 is a transcription factor that regulates mitochondrial dynamics leading to an increase in oxidative phosphorylation (<https://doi.org/10.3389/fnagi.2021.660843>) (**new Figure 4I**).

Figure 4I Western blot showing TOMM20 and CHCHD2 levels, compared to housekeeping β -TUBULIN in hESC-derived oligodendroglial monocultures treated for 7 days with metformin versus vehicle, with quantification. Mean \pm SEM. Paired t-test. M=metformin treated, C=vehicle treated controls.

6. Quantitative data on oligodendroglial markers and myelination at the protein level are lacking. For instance, what percentages of OLIG2+, O4+, and MBP+ cells are observed in monocultures and spheroids? What is the extent of myelination in the spheroid models in vitro?

Data for percentages of oligodendroglia in our monocultures at week 1 and week 2 timepoints are below:

Data for monocultures treated with or without metformin or clemastine shown as fold change but also with percentages with O4 and MBP as differentiation markers (**supplementary figure 1A**) and shown below.

New Figure 2C

Supp Figure 4A & B

Data for organoids showing percentages of the more mature differentiation markers CC1⁺ oligodendrocytes and MBP⁺ oligodendrocytes (similar as expected) and extent of myelination (myelin area) in untreated (blue) and treated (pink) organoids (new **Figure 2C** and **Supplementary Figure 4A,B**) and below.

7. The authors do not provide direct evidence that metformin promotes myelination by transplanted human cells.

We have included data from metformin (and water control) treatment of *Shiverer/Rag2^{-/-}* mice without cell transplants, showing that compact myelin is not present. This confirms that mouse shiverer oligodendrocytes do not respond to metformin by making compact myelin. Therefore, the enhanced compact myelination in the transplanted mice comes from the human oligodendrocytes (**Supplementary figure 7B**, and below).

This complements our finding of increased myelinated axons (**Figure 3C**). We have also altered our language in response to another reviewer's comments, as we only included compact myelin profiles in our quantification. Other non-compacted myelin profiles were grouped with axons with no myelin and we have clarified this in the text.

Specific comments:

1. The rationale behind the treatment schemes for monocultures (7 days) and organoids (days 60–70) is unclear. Please clarify the exact timing of treatments and whether the chosen metformin concentration (100 μ M) is achievable in the human CNS. Were dose-response curves or alternative incubation periods tested?

We apologise that this was unclear. We have now explained this better in the methods, reproduced below.

Monocultures were treated every 2 days from DIV65 to DIV72 with either clemastine fumarate (1 μ M; SML0445-100MG, Scientific Laboratory Supplies Ltd.) or metformin hydrochloride (100 μ M; PHR1084-500MG, Scientific Laboratory Supplies Ltd.), or their respective vehicles, DMSO and ddH₂O, for 7 days (**Figure 1A**). Cortical brain organoids were treated daily between DIV60 to DIV70 with metformin hydrochloride (100 μ M; PHR1084-500MG, Scientific Laboratory Supplies Ltd.) or ddH₂O for 10 days (**Figure 2A**). The organoids were then differentiated further in drug-free medium until DIV 100. Chimeric animals were treated daily with metformin hydrochloride (300mg/kg; PHR1084-500MG, Scientific Laboratory Supplies Ltd.) or ddH₂O by oral gavage for 21days from 92DPT to 113DPT. Treatment was initiated at 42DPT. These doses have previously been used by ourselves³ and others⁴, at timepoints pertinent to affect differentiation.

2. Figure 1D and E: Does the distribution of oligodendroglial subsets based on RNA align with protein-level findings?

We have compared immunohistochemistry with OLIG2 antibody and scRNAseq levels of OLIG2 over time (week1 to week 3) in the monocultures, and found that the direction of change is similar over time, but the percentage of cells expressing the RNA is significantly higher than the cells detected as expressing the protein.

Of these OLIG2⁺ cells, we have compared the protein and scRNAseq data for PDGFRA, GPR17 and MBP, as below.

There are differences in proportions, perhaps as expected as RNA transcripts are expressed before proteins and RNAseq is more sensitive than antibody staining. However, that said, there are similarities, with a reduction of oligodendroglial cells over time in both – consistent with the poorer health of week 3 cultures and astroglial increase - and with an increased proportion of immature PDGFRA⁺ cells compared to more mature MBP⁺ cells in both. This emphasises the immaturity of monocultures of hESC-derived oligodendroglia.

4. Figure 2B: Scale bars are missing. Please include vehicle controls for comparison.

Thank you. We have added the missing scale bar.

These are simply representative images of organoid cultures, not treated with metformin or vehicle.

5. Figure 2C: MBP density might indicate changes in either the number of MBP⁺ cells or the size/branching of individual cells. Could the authors provide data on the proportion of MBP⁺ cells among total cells (or SOX10⁺ oligodendroglial cells)?

This was also asked for by the other reviewers and we agree that it is important.

We have addressed this using immunohistochemistry using antibodies to CC1 and MBP to quantify the number of cell bodies and comparing this to the MBP area signal (relates to processes and myelin sheaths). This shows that metformin treatment of our human organoids does not increase the number of mature CC1+ or MBP+ oligodendrocytes (a minority of more mature MBP+ oligodendrocytes can be CC1-, as shown by these data). However, metformin treatment does increase MBP area, suggesting an increase in myelin production per cell rather than an increase in the number of cells. These data are added to **Supplementary Figure 4A,B** (and shown below).

New Figure 2C

Supp Figure 4A & B

6. Figures 2D and E: Only 301 of ~12,000 cells belong to the oligodendroglial lineage, with a small fraction in the mature Oligo2 subcluster. Are these findings consistent with protein data? If not, this discrepancy may highlight limitations of the scRNAseq approach. If yes, this may suggest that the spheroid model requires further optimization to enhance oligodendroglial differentiation.

We have compared our data with that of other published organoid data. Data from the Pasca lab who pioneered CNS organoids⁵ found ~9% of cells were OLIG2+ oligodendroglia and there were about ~5 MBP+ cells/mm² detected at a similar time point (at the protein level). A later paper focusing on myelin production in CNS organoids⁶ using a similar protocol showed ~11% of cells in myelinoids were SOX10+ (at the protein level). Our level of ~3% is in keeping with this, especially bearing in mind that cells are heterogeneously distributed throughout these organoids. However, as the reviewer indicates, this culture was not ideal for us to answer the biological question that we posed due to the restricted number of oligodendroglia and their maturation. This is why we turned to the chimera model.

7. **Figure 2G:** The Oligo2 subcluster shows general similarity to various in vivo subclusters but no strong alignment with any specific one. Could the authors discuss whether this result was expected and what does this finding mean?

This observed general similarity between the co_Oligo_2 subcluster and various oligodendrocyte subclusters in the post-mortem dataset, without a strong alignment to any specific subcluster, is likely due to the limited number of oligodendrocytes present in the organoids. This restricted population does not fully capture the heterogeneity seen in vivo, where distinct oligodendrocyte subtypes arise in response to diverse developmental and environmental cues. We believe that this is a limitation of organoids for studying oligodendrocyte biology.

As the cosine similarity plot primarily reflects shared gene expression patterns, the alignment we observe likely corresponds to fundamental oligodendrocyte functions, such as myelin genes, conserved across all oligodendrocyte subtypes.

8. Figure 3B: Please provide comparisons to non-transplanted controls and include double-staining for huNu and MBP. If each dot represents an animal, specify this in the figure legend.

We have now included electron microscopy images for metformin (and ddH₂O control) treatment of *Shiverer/Rag2^{-/-}* mice without cell transplants (**Supplementary figure 7B**, and below).

We have provided immunofluorescence pictures of chimeric transplants (including HuNu, GFP and MBP) in **supplementary figure 5C,D**.

For **Figure 3B**, we have added that each dot represents an animal to the legend as requested.

9. **Figure 3C**: How do the authors attribute increased myelinated axons specifically to human cells? Since shiverer mice exhibit loose myelin wrapping, can the authors quantify MBP+ oligodendrocytes or axons with compact myelin in transplanted mice treated with vehicle versus metformin? Is there any change in MBP/NF colocalization in metformin treated animals?

We agree that shiverer mouse oligodendrocytes can have uncompacted loose wrappings of membranes. However, we only included compact myelin profiles in our quantification which are by definition from human oligodendrocytes and have altered our language to make this clearer. Other non-compacted myelin profiles were grouped with axons with no myelin and we have clarified this in the text. We show non-transplanted mouse electron micrograph pictures for reference (**Supplementary figure 7B**).

We also carried out immunofluorescence to determine the number of mature (CC1+) and human (HuNu+) oligodendrocytes in the chimeric animals – as shown below and in **Supplementary figure 5E,F**.

Supplementary Figure 5:

(E) Percentage of all cells which are OLIG2+HuNu+CC1+ oligodendrocytes, and (F) of all human oligodendroglia which are CC1+ in the chimeric mouse corpus callosum 70 days post transplantation comparing control and metformin-treated groups. Points are individual mice, means +/- SEM, p values stated.

We do see a difference in the myelin thickness around axons by electron microscopy, after metformin (**Supplementary figure 5H**) but do not see a change in myelin localisation around axons.

10. Figures 4A and C: How do the authors differentiate between myelin sheaths generated by mouse versus human oligodendrocytes?

The human oligodendrocytes are the only ones that can make compact myelin. Shiverer mouse oligodendrocytes lack functional MBP and so cannot closely appose the myelin membranes as shown below (arrow) (scale bar 0.5um).

Therefore, the human cells in **Figure 4A** are producing the black myelin rings, with more of them in **Figure 4B** when metformin is added. We have added a picture of shiverer mice with no human

cells added as comparison, with and without metformin to make this even clearer (**Supplementary Figure 7B**).

11. Figures 4B and D: The inclusion of pseudo-replicates inflates sample numbers. Please use averages for mitochondrial or myelinated axon area per mouse and compare vehicle-treated versus metformin-treated animals (n=4 and 5, respectively).

These data were in our supplementary figures but we have now moved these to the main **figure 4** as a better location.

12. Figure 4: While metformin upregulates certain genes (e.g., EIF1, NDUFA11, COX8A), this upregulation is not cell-type-specific. Experiments linking these changes to oligodendroglial differentiation or myelination are lacking, limiting the conclusions drawn.

We agree that these changes are not cell-type specific but we actually found this more interesting than the fact that metformin increased myelin production. We have now measured the mitochondrial area in Shiverer mouse axons not surrounded by compact myelin with and without metformin treatment and found the same effect – that metformin increases mitochondrial profiles (now shown in **supplementary figure 7A**).

Supplementary Figure 7A: Significant increase of mitochondrial area in axons without compact myelin (similar to those with myelin) after metformin treatment compared to vehicle-treated controls (average per mouse), means ± SEM.

This shows us that the effect of metformin on mitochondria in axons may not drive the myelination enhancement, nor does myelination cause the mitochondrial effect, but these may be independent. This is interesting to us as it suggests that metformin may be having a direct effect on axons/neurons which may alter metabolism for potential benefit as well as an indirect effect enhancing myelination (and thus neuroprotection). Changes of mitochondrial function-related transcripts in the mouse microglia and astrocytes as well (developed more in the test and in **supplementary figure 9B**, in response to another reviewer's comments, also increases evidence for multiple cellular and pathway effects. This gives us more reason to change the title of this article as suggested.

13. Figure 4: Data from MS tissue samples have low statistical power due to small sample sizes; therefore, the relevance of the findings remains unclear.

We completely recognise that we have only identified a very limited number of people with MS who donated their brains for research who we know were definitely and definitely not taking metformin at death. However, these precious samples are very difficult to find, but having found some, we wanted to include these data as we still found it interesting. We felt we were measured in acknowledging this, but we have made this even more so.

14. Discussion: The authors should address the limitations of their models. For instance, stem cell-derived oligodendrocytes represent “young” cells lacking aging signatures and likely behave differently from adult human OPCs or mature oligodendrocytes (as shown in rodents). Moreover, the models do not replicate inflammatory environments characteristic of MS.

The limitations of the monoculture and organoids are very much one of the points of the paper and so we have re-emphasised even more in the discussion. None of these models replicates any aspect of MS, as we were not testing these in demyelination or inflammation, but this is not what we set out to do. Instead, we were trying to see if human cells would respond to metformin. We have made sure this is clear in the discussion.

References

1. Li, W. *et al.* High-throughput screening for myelination promoting compounds using human stem cell-derived oligodendrocyte progenitor cells. *iScience* **26**, 106156 (2023).
2. Chamling, X. *et al.* Single-cell transcriptomic reveals molecular diversity and developmental heterogeneity of human stem cell-derived oligodendrocyte lineage cells. *Nat Commun* **12**, 652 (2021).
3. Neumann, B. *et al.* Metformin Restores CNS Remyelination Capacity by Rejuvenating Aged Stem Cells. *Cell Stem Cell* **25**, 473-485.e8 (2019).
4. LaMoia, T. E. & Shulman, G. I. Cellular and Molecular Mechanisms of Metformin Action. *Endocrine Reviews* **42**, 77–96 (2021).
5. Marton, R. M. *et al.* Differentiation and maturation of oligodendrocytes in human three-dimensional neural cultures. *Nat Neurosci* **22**, 484–491 (2019).
6. James, O. G. *et al.* iPSC-derived myelinoids to study myelin biology of humans. *Developmental Cell* **56**, 1346-1358.e6 (2021).